# Ultrastrong to nearly deep-strong magnon-magnon coupling with a high degree of freedom in synthetic antiferromagnets

Yuqiang Wang [1,2], Yu Zhang [3], Chaozhong Li[4], Jinwu Wei[4], Bin He[1,2], Hongjun Xu[1,5], Jihao Xia[1,2], Xuming Luo[1,2], Jiahui Li[1,2], Jing Dong[1,5], Wenqing He[1,2], Zhengren Yan[1,2], Wenlong Yang[1,2], Fusheng Ma [3] ✉, Guozhi Chai [4], Peng Yan [6], Caihua Wan [1,2], Xiufeng Han [1,2,5] & Guoqiang Yu [1,2,5] ✉

Ultrastrong and deep-strong coupling are two coupling regimes rich in intriguing physical phenomena. Recently, hybrid magnonic systems have emerged as promising candidates for exploring these regimes, owing to their unique advantages in quantum engineering. However, because of the relatively weak coupling between magnons and other quasiparticles, ultrastrong coupling is predominantly realized at cryogenic temperatures, while deep-strong coupling remains to be explored. In our work, we achieve both theoretical and experimental realization of room-temperature ultrastrong magnon-magnon coupling in synthetic antiferromagnets with intrinsic asymmetry of magnetic anisotropy. Unlike most ultrastrong coupling systems, where the counter-rotating coupling strength $g_2$ is strictly equal to the co-rotating coupling strength $g_1$, our systems allow for highly tunable $g_1$ and $g_2$. This high degree of freedom also enables the realization of normalized $g_1$ or $g_2$ larger than 0.5. Particularly, our experimental findings reveal that the maximum observed $g_1$ is nearly identical to the bare frequency, with $g_1/\omega_0 = 0.963$, indicating a close realization of deep-strong coupling within our hybrid magnonic systems. Our results highlight synthetic antiferromagnets as platforms for exploring unconventional ultrastrong and even deep-strong coupling regimes, facilitating the further exploration of quantum phenomena.

Light-matter interactions have been studied extensively for decades. Traditionally, such studies have focused on single atoms interacting with photons in a cavity[1], where the Hamiltonian can be written in the form of the Jaynes-Cummings model[2] since the coupling strength is much smaller than the bare frequency of the system and the rotating-wave approximation (RWA) is valid. Normalized coupling strength $g/\omega_0$, the ratio of the coupling strength to the bare frequency, is usually used to quantify the regimes of interactions. When $g/\omega_0$ increases to 0.1, the ultrastrong coupling (USC) regime is realized[3,4]. Since the RWA breaks down in this regime, more accurate models are theoretically adopted, such as the quantum Rabi model and Hopfield model[5]. In USC systems, the Hamiltonian of interaction contains not only the co-

[1]Beijing National Laboratory for Condensed Matter Physics, Institute of Physics, Chinese Academy of Sciences, Beijing 100190, China. [2]Center of Materials Science and Optoelectronics Engineering, University of Chinese Academy of Sciences, Beijing 100049, China. [3]Jiangsu Key Laboratory of Opto-Electronic Technology, School of Physics and Technology, Nanjing Normal University, Nanjing 210046, China. [4]Key Laboratory for Magnetism and Magnetic Materials of the Ministry of Education, Lanzhou University, Lanzhou 730000, China. [5]Songshan Lake Materials Laboratory, Dongguan, Guangdong 523808, China. [6]School of Electronic Science and Engineering and State Key Laboratory of Electronic Thin Films and Integrated Devices, University of Electronic Science and Technology of China, Chengdu 610054, China. ✉e-mail: phymafs@njnu.edu.cn; guoqiangyu@iphy.ac.cn

rotating term but also the counter-rotating term, and sometimes the diamagnetic term needs to be considered. The combination of these terms as well as their large values, leads to interesting physical phenomena, such as nontrivial ground state[4] and superradiant phase transition[6,7]. So far, the USC has been widely explored in superconducting circuits[8,9] and semiconductor quantum wells[10–12]. Some studies even show $g/\omega_0$ in excess of 1[13,14], which means the deep-strong coupling (DSC) regime is realized[15]. However, in these traditional USC systems, cryogenic temperature is compulsorily required, and the counter-rotating coupling strength $g_2$ is bound to the co-rotating coupling strength $g_1$, i.e., $g_1 = g_2$, which hinders further study of the USC. Therefore, it is particularly important to look for more systems for further exploration of the USC and DSC regimes.

Hybrid magnonic systems, in which the collective excitations of spins are coupled with other quasiparticles, have been intensively studied in the last decade[16–20]. By utilizing the unique properties of magnon, these systems have been proven to be of great potential in quantum information processing, storage, and sensing[21–23]. Since the dipolar interaction between magnons and photons is inherently weak, only a few works have reported the magnon-photon USC under cryogenic temperature[24–27], and the DSC regime has not been realized yet. In recent years, magnon-magnon interaction in hybrid magnonic systems has been emerging as a new object for exploring coupling phenomena[28–32]. Unlike the magnon-photon coupling, the inherently strong interaction between two magnon modes leads to an enhanced coupling strength. So far, the ultrastrong magnon-magnon coupling has been experimentally reported in a compensated ferrimagnet and antiferromagnets[30,33,34], where the observed maximum normalized coupling strengths are <0.4, without approaching the DSC regime. The realization of the USC regime in these materials usually requires tough experimental conditions, such as cryogenic temperatures or large magnetic fields (>10 T). Besides, the macroscale crystal samples are not compatible with the complementary metal-oxide-semiconductor platforms, which may hinder the practical applications. Alternatively, synthetic antiferromagnets (SAFs) have been demonstrated as good platforms for studying the magnon-magnon coupling at room temperature and with a magnetic field smaller than 1 T[35–39]. In a SAF, the lower ferromagnetic layer (FM1) and upper ferromagnetic layer (FM2) are separated by a spacing non-magnetic layer, which prevents direct exchange interaction but introduces a Ruderman-Kittel-Kasuya-Yosida (RKKY) type interaction[40] between the two FMs. By adjusting the properties of each sublayer individually, the dynamic response of a SAF can be widely tuned[41–43]. Potentially, SAFs could be studied for magnon-magnon interaction in the USC or even the DSC regime. However, a comprehensive investigation is still required.

Here, by introducing an intrinsic asymmetry of magnetic anisotropy between the two FMs, we theoretically and experimentally demonstrate the realization of tunable room temperature magnon-magnon USC in three SAF configurations: perpendicular magnetic anisotropy type (PMA), T-shaped magnetization type (T-Type), and in-plane magnetic anisotropy type (IP). By quantizing our hybrid magnonic systems with the generalized Hopfield model, we demonstrate that the magnon-magnon coupling properties in SAF vary with the magnetization configurations, where the values of $g_1$ and $g_2$ follow different laws. Depending on the specific configurations, the normalized counter-rotating coupling strength $g_2/\omega_0$ can be manipulated as $g_2/\omega_0 = 0$, $g_2/\omega_0 \approx g_1/\omega_0$, or $g_2/\omega_0 \gg g_1/\omega_0$. Such a high degree of freedom of $g_{1(2)}$ can greatly overcome the limitation that $g_{1(2)}/\omega_0$ is not greater than 0.5 caused by the superradiant phase transition and potentially makes the coupling in hybrid magnonic systems toward the DSC regime. Experimentally, we demonstrate the near achievement of the DSC regime with $g_1/\omega_0$ as large as 0.963 in PMA SAF. Our findings suggest that SAFs are ideal platforms for further exploration of quantum phenomena in the USC and even the DSC regime, such as the highly tunable squeezing effect[44,45] of the ground state.

## Results

### Theoretical description of the magnon-magnon coupling in SAFs

In previous works, the magnon-magnon coupling strength is usually approximated as half the size of the gap[30,31]. However, this is not accurate when the coupling strength reaches the USC regime. A detailed example is shown in Supplementary Materials Section S1. Therefore, a quantum model is required in order to correctly quantify the coupling strength. We use the generalized Hopfield Hamiltonian to describe the magnon-magnon hybrid systems[7,33]:

$$\hat{\mathcal{H}} = \hbar\omega_+(\hat{a}_+^\dagger \hat{a}_+ + \tfrac{1}{2}) + \hbar\omega_-(\hat{a}_-^\dagger \hat{a}_- + \tfrac{1}{2}) + \\ i\hbar g_1(\hat{a}_+ \hat{a}_-^\dagger - \hat{a}_+^\dagger \hat{a}_-) + i\hbar g_2(\hat{a}_+^\dagger \hat{a}_-^\dagger - \hat{a}_+ \hat{a}_-) \tag{1}$$

where $\hat{a}_+$ ($\hat{a}_-$) corresponds to the annihilation operator of the + (−) magnon mode, and $\hat{a}_+^\dagger$ ($\hat{a}_-^\dagger$) corresponds to the creation operator of the + (−) magnon mode. Here, "+ magnon mode" and "− magnon mode" refer to different magnon modes. $g_1$ and $g_2$ correspond to the co-rotating coupling strength and counter-rotating coupling strength, respectively. The co-rotating term describes the annihilation (creation) of one + magnon and the creation (annihilation) of one − magnon, while the counter-rotating term describes the annihilation (creation) of one pair of + and − magnons. Different from the normal Hopfield Hamiltonian, $g_2$ here is an independent parameter and does not need to be equal to $g_1$.

Next, we discuss in detail the effects of asymmetry of magnetic anisotropy on coupling properties in SAF systems. We first illustrate how the magnetic anisotropies affect the magnetization states of SAFs. The structure of a SAF is sketched in Fig. 1a. Two ferromagnetic layers: FM1 and FM2, are separated by a spacing non-magnetic layer NM. An external magnetic field **H** is applied in plane, which alters the equilibrium distribution of the two normalized magnetic moments $\mathbf{m_1}$ and $\mathbf{m_2}$. For convenience, we construct a Cartesian coordinate system: $\mathbf{e_z} \equiv \mathbf{n}$, $\mathbf{e_y} \equiv \mathbf{H}/H$, $\mathbf{e_x} \equiv \mathbf{e_y} \times \mathbf{e_z}$, where $H$ refers to the intensity of magnetic field **H**, and **n** refers to the unit vector parallel to the normal of SAF. To quantify the magnetic anisotropies, we define $H_{k1}$ and $H_{k2}$ as the total magnetic anisotropy fields for FM1 and FM2, respectively. $H_k > 0$ refers to perpendicular magnetic anisotropy, and $H_k < 0$ refers to easy-plane magnetic anisotropy. Depending on $H_{k1}$ and $H_{k2}$, the SAF exhibits three kinds of magnetization configurations: IP, PMA, and T-Type, as schematically shown in Fig. 1b, c. The specific method for identifying different SAFs is detailed in Section S2. Dynamically, $\mathbf{m_i}$ ($i = 1, 2$) can precess around its own equilibrium. The static magnetization configurations and dynamic responses are calculated based on the Landau-Lifshitz-Gilbert (LLG) equations, which are shown in detail in Methods and Section S3.

Then, we consider the magnon-magnon coupling properties in our SAF systems. In our systems, "+" and "−" magnon modes represent optical (out-of-phase) and acoustic (in-phase) magnon modes, respectively, as shown in the inset of Fig. 1d. The derivation of Eq. 1 and the specific forms of the parameters are shown in Section S5 and Methods, respectively. In this theoretical section, we show the simplified case, in which we only consider the effects caused by the asymmetry of magnetic anisotropy alone, as detailed in Methods. When $H_{k1}$ deviates from $H_{k2}$, i.e., forming the asymmetry of magnetic anisotropy, a gap is induced, indicating the occurrence of the coupling between the + mode magnon and the − mode magnon, see examples shown in Supplementary Materials Fig. S2. We also discuss the resonance features under different field geometries and different degrees of asymmetry, as shown in Section S4, which directly shows the difference in mode excitation efficiency caused by coupling between the + and − mode magnons. Here, we focus on three intriguing cases corresponding to different magnetization configurations, as shown in Fig. 1d–f, respectively. The color plots in the top panels show the resonance spectra obtained by micromagnetic simulations (see details

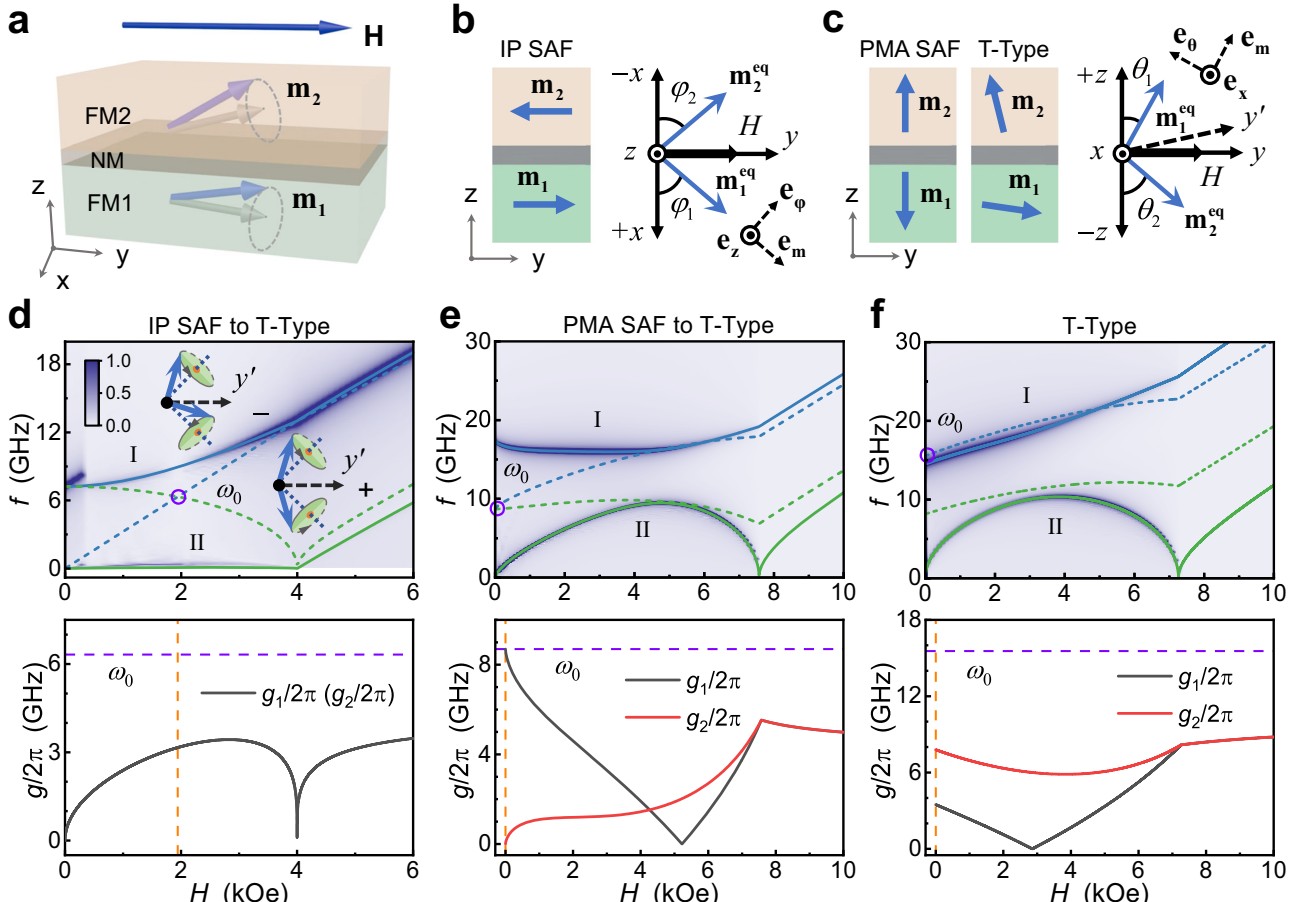

**Fig. 1 | Schematic illustrations of magnetization configurations and coupling properties of synthetic antiferromagnets (SAFs). a** Sketch of a typical SAF. $\mathbf{m}_i$ ($i$ = 1, 2) precesses around its equilibrium axis. **b, c** Sketches of static equilibrium configurations for in-plane magnetic anisotropy type (IP) SAF (**b**), perpendicular magnetic anisotropy type (PMA) SAF and T-shaped magnetization type (T-Type) (**c**) at zero magnetic field $H$. The coordinate systems are introduced for convenience. **d**–**f** Top: theoretically calculated (solid and dashed lines) and simulated (color plots) resonance spectra; bottom: the corresponding coupling strengths $g_1$ and $g_2$. **d** corresponds to a critical IP SAF case, where the total magnetic anisotropy fields $H_{k1}$ = 1.428 kOe, $H_{k2}$ = −5 kOe for theoretical calculation, and the magnetic anisotropy energies (excluding demagnetizing energy) $K_{u1}$ = 2.68 × 10⁵ J/m³,

$K_{u2}$ = 0.76 × 10⁵ J/m³ for micromagnetic simulation. **e** corresponds to a critical PMA SAF case, where $H_{k1}$ = −1.428 kOe, $H_{k2}$ = 5 kOe, and $K_{u1}$ = 1.83 × 10⁵ J/m³, $K_{u2}$ = 3.76 × 10⁵ J/m³. And **f** corresponds to a T-Type case, where $H_{k1}$ = −10 kOe, $H_{k2}$ = 5 kOe, and $K_{u1}$ = −0.74 × 10⁵ J/m³, $K_{u2}$ = 3.76 × 10⁵ J/m³. Gyromagnetic ratio $\gamma/2\pi$ is 2.7 GHz/kOe. In top panels, the blue and green solid curves represent the calculated high-frequency branches (I) and low-frequency branches (II), respectively. The blue and green dashed curves represent the corresponding decoupled − and + modes, respectively, and their schematics are shown in the inset in **d**. Violet circles indicate the center frequencies $\omega_0$. In bottom panels, the orange dashed lines indicate the cases of $H = H_0$, and the violet dashed lines indicate the center frequencies $\omega_0$.

in Methods), which is consistent with the theoretical results (solid curves). The corresponding $g_1/2\pi(H)$ and $g_2/2\pi(H)$ based on Eq. 1 are given in the bottom panels. Unless otherwise specified, the RKKY interlayer coupling field $H_{ex}$ is set to −2 kOe in this section. We begin with the IP SAF, which is the configuration primarily considered in previous magnon-magnon coupling studies in SAFs[35–38]. By selecting the appropriate values of magnetic anisotropies, the IP SAF configuration can be adjusted to a critical case, for instance, $H_{k1}$ = 1.428 kOe and $H_{k2}$ = −5 kOe. In this critical IP SAF case, when the value of $H_{k1}$ increases, the magnetization configuration will change into T-Type. The corresponding resonance and coupling characteristics of this critical IP SAF are shown in Fig. 1d. Interestingly, we find that the frequency of low-frequency branch II $\omega_{II}$ decreases to zero, which means that the gap is opened to the maximum extent. Although a similar zero-frequency branch can be induced by applying a perpendicular **H** to the sample plane, coupling will not occur in this case since the rotation symmetry of the system is preserved[31]. Therefore, in this previous study case, the high-frequency branch I and low-frequency branch II directly correspond to the pure − mode and + mode, respectively. However, in our case, the large gap is entirely caused by the coupling

effect, reflected in the fact that the + mode and − mode intersect at frequency $\omega_0$. This frequency $\omega_0$ corresponds to the center frequency, which indicates the bare energy of the decoupled magnon. In the bottom panel of Fig. 1d, we show that $g_1$ and $g_2$ are functions of $H$, and exhibit significant non monotonicity. Note that in IP SAF, $g_1$ is equal to $g_2$ for any $H$, which is similar to the traditional light-matter hybrid systems. We focus on the coupling strengths at the magnetic field $H_0$ corresponding to the center frequency $\omega_0$, as shown by the dashed vertical line. In the critical IP SAF case, $g_{1(2)}(H_0)$ is equal to half of $\omega_0$, indicating that the IP SAF system is deep into the USC regime. However, due to the superradiant phase transition when $\omega_{II}$ decreases to zero, the maximum $g_1$ and $g_2$ are limited to half of the $\omega_0$, which means the IP SAF system is still far from realizing the DSC regime.

Here, we must point out that $g_1 = g_2$ is only a special situation for our systems. In SAFs, generally, $g_1$ does not necessarily have to be equal to $g_2$, as demonstrated in Methods. Therefore, achieving large $g_1$ or $g_2$ exceeding 0.5 is feasible in various magnetization configurations, despite being limited by the superradiant phase transition. We first illustrate the realization of large $g_1$ in PMA SAF. In this configuration, $g_1 \neq g_2$ when $H$ is less than the saturation field, and the + mode and

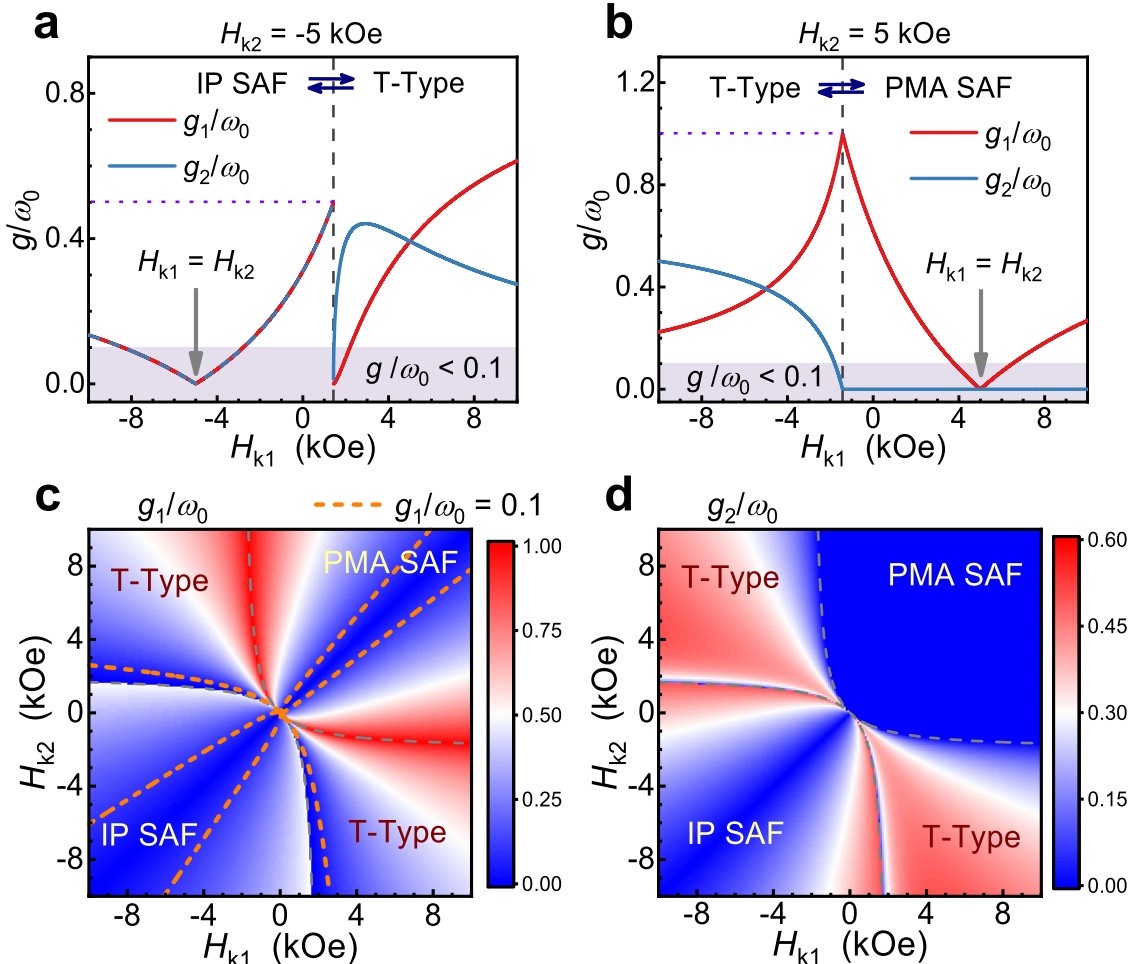

**Fig. 2 | Theoretical calculation of normalized coupling strengths of SAFs.** **a**, **b** The normalized coupling strengths $g_1/\omega_0$ and $g_2/\omega_0$ as functions of $H_{k1}$ when $H_{k2} = -5\,\text{kOe}$ (**a**) and $H_{k2} = 5\,\text{kOe}$ (**b**). The gray dashed lines divide each plot into two parts, corresponding to IP SAF and T-Type in **a**, and PMA SAF and T-Type in **b**. The light purple color marks the region that does not satisfy the ultrastrong coupling (USC) regime. The violet dotted lines indicate the maximum $g_{1(2)}/\omega_0$ that can be achieved in IP SAF (**a**) and PMA SAF (**b**). **c**, **d** Color plots of calculated $g_1/\omega_0$ (**c**) and $g_2/\omega_0$ (**d**) as functions of $H_{k1}$ and $H_{k2}$. The gray dashed curves, which are calculated based on Section S2, divide each color plot into four regions: one PMA SAF region, one IP SAF region and two T-Type regions. The orange dashed curves in **c** represent the contour lines with $g_1/\omega_0 = 0.1$, which indicate the lower limit of USC.

− mode intersect at $H = H_0 = 0$ Oe. Similarly, we consider a critical case for $H_{k1} = -1.428\,\text{kOe}$ and $H_{k2} = 5\,\text{kOe}$. In this critical PMA SAF case, the magnetization configuration will change into T-Type as the value of $H_{k1}$ decreases. The corresponding resonance and coupling characteristics of this critical PMA SAF are shown in Fig. 1e. In this case, $\omega_{\text{II}}$ decreases to zero at $H = H_0 = 0$ Oe, which is similar to Fig. 1d. However, differently, the gap at $H_0$ is exactly twice as large as $\omega_0/2\pi$ in this case, indicating the elimination of the counter-rotating coupling when $H = H_0$. Intriguingly, this also implies the existence of a $g_1(H_0)$ equal to $\omega_0$, as illustrated in the corresponding bottom panel. Such a large $g_1(H_0)$ in PMA SAF even reaches the lower limit of the DSC regime, which has only been realized in a few hybrid systems[13,14,46].

Next, we illustrate the realization of large $g_2$, which is achieved in T-Type. Figure 1f shows a typical T-Type case with $H_{k1} = -10\,\text{kOe}$ and $H_{k2} = 5\,\text{kOe}$. Although, in this case, $\omega_{\text{II}}$ remains zero at $H = 0$ Oe, which is the same as that in Fig. 1e, the coupling properties are completely different from the PMA SAF. In the T-Type case, $\omega_{\text{I}}$ is even smaller than the − mode frequency $\omega_-$ when $H$ is small, which is a feature of the vacuum Bloch-Siegert shifts (VBSSs)[33,47]. The VBSSs imply the existence of a large $g_2$, as illustrated in the bottom panel of Fig. 1f, representing a stark contrast to the PMA SAF. In this case, a significant feature is the absence of an intersection between the − mode and + mode. While previous studies have seldom addressed such a situation because the

gap caused by the coupling only contributes to a portion of the total gap, the quantum model defines $g_1$ and $g_2$ at any $H$, making them meaningful for the T-Type case. To normalize the coupling strengths, we also define $\omega_0$ in the T-Type case. To avoid exaggerating the effect of coupling and considering that T-Type and PMA SAF can be described within the same model, we set $H_0$ to zero, and $\omega_0$ is determined as the larger value between $\omega_+(H_0)$ and $\omega_-(H_0)$. In our cases, $\omega_0$ is determined as $\omega_-(H_0)$. Notably, $g_2(H_0)$ exceeds half of $\omega_0$, and is also larger than twice $g_1(H_0)$. By applying a certain $H$, $g_1$ can even be reduced to 0, while $g_2$ remains at a substantial value, indicating the possibility of realizing a large pure counter-rotating coupling in the T-Type configuration.

The aforementioned results demonstrate the capability of SAFs to realize the USC regime and even reach the low limit of the DSC regime. To further illustrate the tunability of coupling, here, we focus on the coupling properties at $H = H_0$. We calculate the normalized coupling strengths $g_{1,2}(H_0)/\omega_0$, denoted as $g_{1,2}/\omega_0$ for simplicity in the subsequent discussion. Notably, for the IP SAF and PMA SAF, $g_1/\omega_0$ and $g_2/\omega_0$ have straightforward analytical expressions, as detailed in Methods. As an example, Fig. 2a, b show $g_1/\omega_0$ and $g_2/\omega_0$ as functions of the magnetic anisotropy field $H_{k1}$, with $H_{k2}$ fixed at $-5\,\text{kOe}$ (Fig. 2a) and $5\,\text{kOe}$ (Fig. 2b). The gray dashed lines, corresponding to the critical cases mentioned in Fig. 1, divide each plot into two parts with different

configurations, i.e., T-Type and IP SAF in Fig. 2a, and PMA SAF and T-Type in Fig. 2b. We note that different ranges of $g_1/\omega_0$ and $g_2/\omega_0$ can be realized in these three configurations. In IP SAF, $g_2/\omega_0$ equals $g_1/\omega_0$. In PMA SAF, $g_2/\omega_0$ equals 0, independent of the value of $g_1/\omega_0$, indicating the strict validity of the RWA even in the USC regime. While in T-Type, $g_2/\omega_0$ exhibits significant flexibility. Depending on the specific values of parameters, $g_2/\omega_0$ can be less than, equal to, or larger than $g_1/\omega_0$. Furthermore, the normalized coupling strengths can be continuously tuned in each SAF region. In the IP SAF and PMA SAF regions, with the increase of $|H_{k1}-H_{k2}|$, $g_1/\omega_0$ increases monotonically from 0 to more than 0.1. At the boundaries of the regions, i.e., the cases indicated by the gray dashed lines, $g_1/\omega_0$ reaches its maximum: 0.5 for IP SAF and 1 for PMA SAF. When the degree of asymmetry further increases, the configuration transforms into T-Type in each plot. $g_1/\omega_0$ and $g_2/\omega_0$ change continuously during the transition from PMA SAF to T-Type. However, when the configuration transforms from IP SAF to T-Type, due to the abrupt change in equilibrium positions of $\mathbf{m_1}$ and $\mathbf{m_2}$, $g_1/\omega_0$ and $g_2/\omega_0$ exhibit sudden and discontinuous decreases.

In order to provide a more comprehensive understanding of the relationship between the normalized coupling strengths and the anisotropic asymmetry, color plots of calculated $g_1/\omega_0$ and $g_2/\omega_0$ are presented in Fig. 2c, d, respectively. These plots illustrate the influence of tuning $H_{k1}$ and $H_{k2}$, resulting in four distinct regions: one PMA SAF region, one IP SAF region, and two T-Type regions, as divided by the gray dashed curves. It is important to note that the above discussions regarding Fig. 2a, b remain applicable when both $H_{k1}$ and $H_{k2}$ change. The orange dotted curves in Fig. 2c represent the lower limit of the USC. It can be seen that the USC regime can be realized across most regions of the plot, indicating the effectiveness of realizing USC by constructing SAF with appropriate magnetic anisotropic asymmetry. As shown in Fig. 2d, the maximum $g_2/\omega_0$ is obtained in T-Type regions, reaching ~0.6, when $H_{ex} = -2$ kOe.

We also investigate the effect of RKKY interaction intensity on $g_1/\omega_0$ and $g_2/\omega_0$, as shown in Section S6. The results show that most of the properties mentioned above hold true when $H_{ex}$ changes, and $H_{ex}$ only plays a minor role in changing the value of $g_1/\omega_0$ and $g_2/\omega_0$. This is because the pure + mode and − mode are combined modes of FM1 and FM2, while the RKKY interaction acts between FM1 and FM2. Therefore, the RKKY interaction does not directly influence the coupling between the + mode and − mode. In addition, a recent study indicates that for a PMA SAF system with a magnetic field perpendicular to the sample plane, its resonance features are closely related to the initial configurations, i.e., head-to-head and tail-to-tail configurations[48]. We demonstrate that in our PMA SAF and T-Type systems, their resonance features are independent of the initial configurations when $H$ is applied in the SAF plane, as shown in Section S10.

## Experimental realization of the USC in SAFs

In this section, we experimentally demonstrate the large coupling strengths in SAFs with asymmetry of magnetic anisotropy. Samples with all three configurations: IP SAF, PMA SAF, and T-Type, are prepared. We use Co/Ni stack as the ferromagnetic layer because of its tunable magnetic anisotropy and relatively small damping[49–51]. An Ir layer is used as the intermediate non-magnetic layer[52]. After preparation, we use the spin torque ferromagnetic resonance (ST-FMR) technique[53–55] to study the magnon-magnon coupling properties of these devices. All the measurements are taken at room temperature unless otherwise specified. Detailed sample structures, the preparation process, and the ST-FMR measurement are displayed in Methods. Figure 3a shows the schematic diagram of the ST-FMR measurement setup. While previous studies mainly use the vector network analyzer (VNA) FMR technique to measure magnon-magnon coupling, we adopt the ST-FMR technique because of its high sensitivity[56,57] and device miniaturization down to the sub-micron scale, as shown in Section S15. Figure 3b shows an example of fitting a typical resonance spectrum

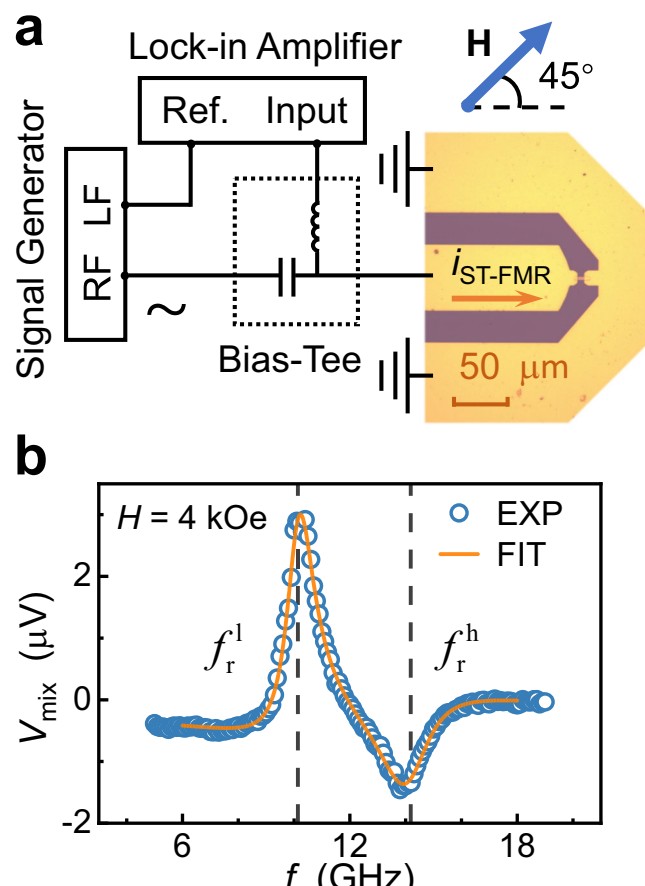

**Fig. 3 | Spin torque ferromagnetic resonance (ST-FMR) measurement setup and measured signal. a** Schematic diagram of setup for ST-FMR measurement. The effective area of the rectangular strip is $4 \times 10\ \mu m^2$. Orange arrow indicates the rf current $i_{STFMR}$ generated by signal generator. Lock-in amplifier is used to detect the generated mixing voltage $V_{mix}$. $H$ is exerted 45° from $i_{STFMR}$ in the sample plane. **b** A typical example of the measured signal of sample S1. The mixing voltage $V_{mix}$ is recorded when scanning the $i_{STFMR}$ frequency $f$ with a fixed $H$ of 4 kOe, as represented by the blue circles. Orange curve is the corresponding Lorentzian fitting result. Vertical dashed lines indicate the extracted resonance frequencies $f_r^l$ and $f_r^h$, respectively.

obtained from the frequency-sweeping ST-FMR. The orange curve corresponds to the fitting result, which originates from Eq. 10 shown in Methods. By fitting the spectra, the resonance frequencies can be extracted, as shown in the dashed lines. In addition to the ST-FMR measurements, as a supplement, we also perform the Brillouin light scattering (BLS) measurements to obtain the resonance spectra[58]. The setup of the BLS measurement is shown in Section S13.

In Fig. 4, we display the experimental results. Figure 4a–c shows the resonance spectra of three typical samples: S3, S1, and S2, corresponding to the IP SAF, PMA SAF, and T-Type configurations, respectively. More details of the spectra are shown in Section S14. The configuration of each sample is verified by the vibrating sample magnetometry (VSM) measurement, which is displayed in Fig. S15. In Fig. 4a–c, the color plots represent the resonance spectra obtained from the ST-FMR measurements, and the red triangles represent the resonance frequencies extracted from the Stokes peaks measured by the BLS technique. We note that two distinct branches appear in each plot, which corresponds to branch I and branch II. The black and red dashed curves represent the theoretically calculated branch I and branch II, respectively, which are obtained by using the material parameters fitted from the resonance spectra. The black and red dotted curves represent the corresponding − mode and + mode,

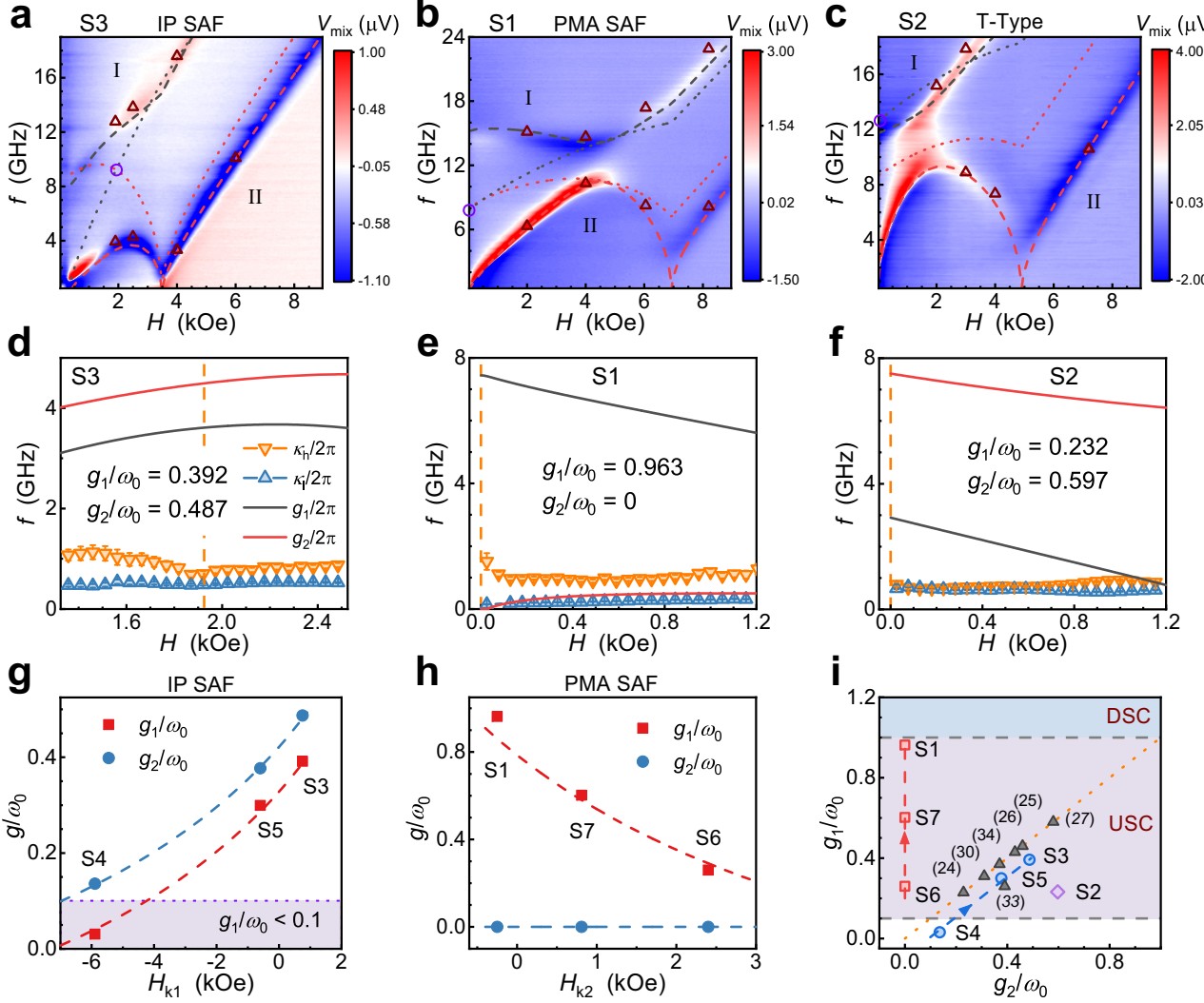

**Fig. 4 | Measured magnon-magnon coupling properties of SAF samples with different magnetization configurations. a–c** Measured resonance spectra for the IP SAF sample S3 (**a**), the PMA SAF sample S1 (**b**) and the T-Type sample S2 (**c**). Color plots show $V_{mix}$ as functions of $H$ and frequency $f$, where color scales indicate the amplitudes. And the red triangles represent the resonance frequencies extracted from the Brillouin light scattering (BLS) Stokes peaks. In each plot, the gray (red) dashed curve represent the fitted branch I (II). And the gray (red) dotted curve represent the corresponding decoupled − (+) mode. The violet circle indicates $\omega_0$. **d–f** Extracted dissipation rates $\kappa_h$, $\kappa_l$ and coupling strengths $g_1$, $g_2$ as functions of $H$ for samples S3 (**d**), S1 (**e**), and S2 (**f**). In each plot, the orange (blue) triangles represent $\kappa_h$ ($\kappa_l$), and dark (red) curve represents $g_1$ ($g_2$). Error bars show the

standard deviations of $\kappa_{h(l)}$. The orange dashed line indicates $H_0$. **g** and **h** Extracted $g_1/\omega_0$ (red squares) and $g_2/\omega_0$ (blue dots) of the three IP SAF samples S3, S4, and S5 (**g**) and the three PMA SAF samples S1, S6, and S7 (**h**). Abscissas record the corresponding $H_{k1}$ or $H_{k2}$. Red (blue) dashed curve in each plot corresponds to the calculated $g_{1(2)}/\omega_0$-$H_k$. The light purple color in **g** marks the region that does not satisfy the USC regime. **i** Comparison of $g_1/\omega_0$ and $g_2/\omega_0$ in our PMA SAF samples (squares), IP SAF samples (circles), T-Type sample (diamond) and some other works (triangles). The numbers in parentheses represent reference numbers. The blue and red dashed curves are derived from (**g, h**). And the orange dotted line indicates the general case with $g_2/\omega_0 = g_1/\omega_0$. The light purple and light blue mark the USC and deep-strong coupling (DSC) regions, respectively.

respectively. Here, to match the actual situation, the complete case is adopted in the fitting, where the asymmetry of magnetic moments of FM1 and FM2, as well as the biquadratic exchange interaction between FM1 and FM2[59] is considered, see details in Methods. While these two factors are taken into account, most of the properties proposed in the theoretical section remain unchanged. The only point to mention is that in IP SAF, $g_2$ is not necessarily equal to $g_1$ in the complete case. The reliability of the theoretical calculations is verified by the coincidence between the sample parameters obtained from fitting the resonance spectra and the parameters obtained from fitting the hysteresis loops, as detailed in Section S12. In all three samples, large magnetic anisotropic asymmetries are formed. We use the generalized Hopfield model to extract the coupling strengths corresponding to the spectra fitting curves. In Fig. 4d–f, we present the corresponding $g_1$, $g_2$ and dissipation rates $\kappa_h$, $\kappa_l$ in the vicinity of $H = H_0$. The result shows that

the USC regime is realized in all three SAF samples. For IP SAF sample S3, the normalized coupling strengths $g_1/\omega_0$ and $g_2/\omega_0$ are determined to be 0.392 and 0.487, respectively. This sample is prepared to approach a critical IP SAF case, except that $g_1/\omega_0$ and $g_2/\omega_0$ are not equal. For PMA SAF sample S1, $g_1/\omega_0$ and $g_2/\omega_0$ are determined to be 0.963 and 0, respectively. It is worth mentioning that such a large $g_1/\omega_0$ approaches the theoretical maximum and almost reaches the DSC regime. The zero $g_2/\omega_0$ in this PMA SAF sample is also consistent with the theoretical section. For T-Type sample S2, $g_1/\omega_0$ and $g_2/\omega_0$ are determined to be 0.232 and 0.597, respectively. Similar to the theoretical case shown in Fig. 1f, we realize a large $g_2/\omega_0$ in this sample, which is around 2.5 times of $g_1/\omega_0$. We note that $g_1/\omega_0$ and $g_2/\omega_0$ extracted from the spectra here are consistent with $g_1/\omega_0$ and $g_2/\omega_0$ calculated from the parameters obtained from fitting the hysteresis loops, as shown in Table S1, indicating the consistency of various

methods in extracting parameters. We also calculate the cooperativities[31], where we define the co-rotating (counter-rotating) cooperativity $C_{1(2)}(H) = g_{1(2)}^2(H)/(\kappa_h(H) \times \kappa_l(H))$ in our systems. Focusing on the case of $H = H_0$, for samples S3, S1, and S2, $C_1(H_0)$ are determined to be 38.3, 249.5, and 16.1, respectively. And $C_2(H_0)$ is determined to be 59.4, 0, and 106.1, respectively. The large values of $C_1(H_0)$ indicate that our systems can effectively transmit information before dissipation. Besides, in order to clarify how the biquadratic exchange interaction and the asymmetry of magnetic moments affect the coupling strengths in our samples, we study the relationship between $g_{1(2)}/\omega_0$ and these two factors, see details in Section S11. The result indicates that although these two factors can affect the coupling strengths, they play a minor role in inducing USC in our samples compared to the asymmetry of magnetic anisotropy.

As shown in Fig. 2, we have theoretically shown that $g_1/\omega_0$ and $g_2/\omega_0$ can be greatly tuned by tuning the anisotropic asymmetry. Here, corresponding experiments are performed to verify this tunability. Taking IP SAF as an example, another two IP SAF samples are prepared: S4 and S5, where the composition and period of the Co/Ni stack corresponding to FM1 are modified, and the rest of the structures are kept the same as S3. Therefore, $H_{k1}$ can be tuned while keeping the other parameters as constant as possible. The resonance spectra as well as the fitting results for samples S4 and S5, are shown in Fig. S18a, b, respectively. The extracted $g_1/\omega_0$ and $g_2/\omega_0$ of the three IP SAF samples are displayed in Fig. 4g. Red (blue) dashed curve corresponds to the $g_{1(2)}/\omega_0$-$H_{k1}$ dependence calculated with a uniform set of parameters. With the increase of $H_{k1}$, the degree of asymmetry between $H_{k1}$ and $H_{k2}$ increases, resulting in enhanced $g_1/\omega_0$ and $g_2/\omega_0$. We note that $g_1/\omega_0$ can be tuned from less than 0.1 (sample S4) to larger than 0.1 (samples S5 and S3), indicating the realization of controlling the USC regime on or off by tuning the degree of magnetic asymmetry. Besides, we prepare another two PMA SAF samples: S6 and S7, where the composition of the Co/Ni stack corresponding to FM2 is modified, and the rest of the structures are kept the same as sample S1. Similarly, by fitting the resonance spectra shown in Fig. S18c, d, $g_1/\omega_0$ and $g_2/\omega_0$ of these two samples are extracted and displayed in Fig. 4h, which also verifies the great tunability of coupling strengths in our SAF systems. Figure 4i compares the coupling strengths in different systems, including the SAF systems in our study and some other USC hybrid magnonic systems[24–27,30,33,34]. An interesting feature of our SAFs is the relative independence between $g_1/\omega_0$ and $g_2/\omega_0$, while in most hybrid magnonic systems, $g_2/\omega_0$ is considered to be equal to $g_1/\omega_0$. Therefore, our SAF systems expand the research scope of coupling phenomena. In addition to tuning the anisotropic asymmetry by modifying the structures of FMs, we also demonstrate the tunability of the asymmetry by tuning the temperature. An example is displayed in Section S16, where we show that a large modification of $g_1$ can be realized.

## Discussion

Our theoretical analysis and experimental measurements show the entirely different ranges of $g_1/\omega_0$ and $g_2/\omega_0$ in different SAFs: IP SAF can serve as a platform with relatively normal USC properties similar to the traditional light-matter coupling; PMA SAF is an ideal platform to study the phenomena caused only by co-rotating coupling, such as the vacuum Rabi splitting-induced shifts (VRSSs); T-Type is a platform for studying the effects caused by different range of counter-rotating coupling, especially the effects caused by large counter-rotating coupling. This high degree of freedom of coupling indicates rich quantum properties in SAFs, such as the squeezing effect of the ground state. As discussed in detail in Sections S7 and S8, we point out that in IP SAF, since $g_2 \neq 0$, the ground state of the Hamiltonian Eq. 1 is a nontrivial ground state, which is a squeezed vacuum of the + mode and − mode magnon, resulting in a reduced quantum fluctuation and a non-zero magnon number. While in PMA SAF, due to the totally co-rotating coupling at $H = H_0$, the ground state of PMA SAF at $H = H_0$ is a trivial

ground state without a squeezing effect. In T-Type, interestingly, a perfect squeezing is obtained when $H$ approaches $H_0$. Such perfect squeezing may increase the sensitivity of quantum measurement and lead to entanglement properties[45,60], which need further exploration. Near the boundaries between different SAF configurations, abrupt changes of the squeezing effect can be obtained by slightly tuning the asymmetry of magnetic anisotropy, which is also an interesting feature of our systems. In addition to the squeezing effect, we also discuss VRSSs, VBSSs, and ground-state energy[61] in our anisotropic asymmetric systems (see details in Section S9). Therefore, the high tunability of coupling in our SAF systems can lead to the high tunability of various quantum phenomena. Understanding these quantum properties will help the applications of SAFs in future quantum technology.

In summary, we theoretically and experimentally demonstrate the effectiveness of magnetic anisotropic asymmetry in SAFs for inducing tunable room temperature ultrastrong magnon-magnon coupling. By quantizing our SAF systems, we obtain the coupling strengths $g_1$ and $g_2$, revealing highly tunable coupling behaviors in three different magnetization configurations. In IP SAF, when the asymmetric factors beyond the magnetic anisotropy are weak, $g_2/\omega_0$ is roughly equal to $g_1/\omega_0$, leading to an experimentally observed maximum $g_{1(2)}/\omega_0$ approaching 0.5. In PMA SAF, $g_2/\omega_0$ equals 0, while a large $g_1/\omega_0$ of 0.963 is experimentally observed, indicating the near-realization of the DSC regime. In T-Type, a dominant $g_2/\omega_0$ near 0.6 is observed, presenting opportunities for studying phenomena caused by the counter-rotating coupling, such as large VBSSs. These results underscore SAFs as appealing systems for further exploration of the USC and even DSC regimes.

## Methods
### Classical method
The static magnetization configurations and dynamic responses are calculated based on the classical method. In a complete model, based on the macrospin approach, the total energy per unit area of a SAF $E$ can be written as

$$
\begin{aligned}
E &= E_{\text{Zeeman}} + E_{\text{Ani}} + E_{\text{Exchange}} \\
E_{\text{Zeeman}} &= -\mu_0 d_1 M_{S1} \mathbf{m_1} \cdot \mathbf{H} - \mu_0 d_2 M_{S2} \mathbf{m_2} \cdot \mathbf{H} \\
E_{\text{Ani}} &= -\frac{1}{2}\mu_0 H_{k1} M_{S1} d_1 (\mathbf{m_1} \cdot \mathbf{e_z})^2 - \frac{1}{2}\mu_0 H_{k2} M_{S2} d_2 (\mathbf{m_2} \cdot \mathbf{e_z})^2 \\
E_{\text{Exchange}} &= -J_1 (\mathbf{m_1} \cdot \mathbf{m_2}) - J_2 (\mathbf{m_1} \cdot \mathbf{m_2})^2
\end{aligned}
\tag{2}
$$

where the $E_{\text{Zeeman}}$ term, the $E_{\text{Ani}}$ term, and the $E_{\text{Exchange}}$ term correspond to Zeeman energy, anisotropy energy, and exchange interaction energy, respectively. $M_{Si}$ and $d_i$ ($i = 1, 2$) refer to the saturation magnetization and the thickness of FM layer $i$, respectively. $J_1$ and $J_2$ correspond to bilinear and biquadratic interlayer exchange coefficients, respectively. For convenience, we define the following parameters: $H_{\text{ex1}}^{(1)} = J_1/\mu_0 d_1 M_{S1}$, $H_{\text{ex1}}^{(2)} = J_1/\mu_0 d_2 M_{S2}$, $H_{\text{ex2}}^{(1)} = J_2/\mu_0 d_1 M_{S1}$, $H_{\text{ex2}}^{(2)} = J_2/\mu_0 d_2 M_{S2}$, which can be understood as the bilinear and biquadratic interlayer coupling fields for each FM. $\mu_0$ is permeability of vacuum. In the theoretical section, however, to highlight the coupling effects caused by the asymmetry of magnetic anisotropy, we consider a simplified case, where we assume FM1 and FM2 share the same saturation magnetization $M_S$ and thickness $d$ and set $J_2$ to be zero. We define a uniform interlayer coupling field: $H_{\text{ex}} = H_{\text{ex1}}^{(1)} = H_{\text{ex1}}^{(2)} = J_1/\mu_0 d M_S$ in this simplified case. The LLG equations are used to solve the equilibrium positions and the dynamic responses of $\mathbf{m_1}$ and $\mathbf{m_2}$ in both complete and simplified cases:

$$
\begin{aligned}
\frac{d\mathbf{m_1}}{dt} &= -\gamma \mathbf{m_1} \times \mathbf{H_{\text{eff}1}} + \alpha_1 \mathbf{m_1} \times \frac{d\mathbf{m_1}}{dt} \\
\frac{d\mathbf{m_2}}{dt} &= -\gamma \mathbf{m_2} \times \mathbf{H_{\text{eff}2}} + \alpha_2 \mathbf{m_2} \times \frac{d\mathbf{m_2}}{dt}
\end{aligned}
\tag{3}
$$

where $\gamma$ is gyromagnetic ratio. $\alpha_1$ and $\alpha_2$ correspond to Gilbert damping constants of FM1 and FM2, respectively. Since $\alpha$ is usually several orders of magnitude less than one, we neglect $\alpha_1$, $\alpha_2$ for simplicity. $\mathbf{H_{effi}} = -(1/\mu_0 dM_S)(\partial E/\partial \mathbf{m_i})$ corresponds to the effective magnetic field of $\mathbf{m_i}$. By solving Eq. 3, the equilibrium positions and the dynamic responses of magnetic moments can be obtained. $\mathbf{m_i}$ has two parts: $\mathbf{m_i} = \mathbf{m_i^{eq}} + \delta\mathbf{m_i}e^{i\omega t}$, where $\mathbf{m_i^{eq}}$ corresponds to the equilibrium configuration, and $\delta\mathbf{m_i}e^{i\omega t}$ corresponds to the dynamic part with precession frequency $\omega$. The schematic diagrams of the $\mathbf{m_i^{eq}}$ in the IP SAF configuration and the PMA SAF and T-Type configurations are shown in Fig. 1b, c. In the calculation, for IP SAF configuration, the $\mathbf{m_i^{eq}}$ resides in the $x$-$y$ plane when $H$ is applied along $\mathbf{e_y}$. $\varphi_1$ ($\varphi_2$) is defined as the angle at which $\mathbf{m_1^{eq}}$ ($\mathbf{m_2^{eq}}$) deviates anti-clockwise (clockwise) from the $+x$ ($-x$) axis. For either PMA SAF or T-Type configurations, the $\mathbf{m_i^{eq}}$ resides in $y$-$z$ plane. $\theta_1$ ($\theta_2$) is defined as the angle at which $\mathbf{m_1^{eq}}$ ($\mathbf{m_2^{eq}}$) deviates clockwise (anti-clockwise) from the $+z$ ($-z$) axis. We note that the equilibrium angles $\theta_{1(2)}$ and $\varphi_{1(2)}$ have only numerical solutions in the complete case, as shown in Section S3. While in the simplified case, $\varphi_1$ ($\varphi_2$) can be solved analytically:

$$\varphi_1(H) = \varphi_2(H) = \begin{cases} \arcsin(H/-2H_{ex}), & H < -2H_{ex} \\ \pi/2, & H \geq -2H_{ex} \end{cases} \quad (4)$$

Note that $\varphi_1$ ($\varphi_2$) is independent of $H_{k1}$ ($H_{k2}$).

Next, we focus on the dynamic responses of the three SAF configurations. Since the phase difference between $\mathbf{m_1}$ and $\mathbf{m_2}$ is constant when precession is stable, we consider the joint precession modes of $\mathbf{m_1}$ and $\mathbf{m_2}$. As shown in Fig. 1b, c, we first define a new $y'$ axis: $\mathbf{e_{y'}} \parallel \mathbf{m_1^{eq}} + \mathbf{m_2^{eq}}$ and an operator $C_{2y'}$. The effect of $C_{2y'}$ is to rotate vectors 180° about the $y'$ axis. In the simplified IP SAF case, $\mathbf{e_{y'}} = \mathbf{e_y}$. Then, two joint precession vectors, $\delta\mathbf{m_+} = \delta\mathbf{m_1} + C_{2y'}\delta\mathbf{m_2}$ and $\delta\mathbf{m_-} = \delta\mathbf{m_1} - C_{2y'}\delta\mathbf{m_2}$, are introduced, which correspond to pure optical (+) mode and pure acoustic (−) mode, respectively. Under the joint operations of $C_{2y'}$ and lattice exchange, the + (−) mode possesses even (odd) parity. Then we define two local coordinate systems, $\mathbf{e_m} \parallel \mathbf{m_1^{eq}}$, $\mathbf{e_z}$ and $\mathbf{e_\varphi} \parallel \mathbf{e_z} \times \mathbf{e_m}$ for IP SAF, and $\mathbf{e_m} \parallel \mathbf{m_1^{eq}}$, $\mathbf{e_x}$ and $\mathbf{e_\theta} \parallel \mathbf{e_x} \times \mathbf{e_m}$ for PMA SAF (T-Type). $\delta\mathbf{m_+}$ and $\delta\mathbf{m_-}$ are projected onto these new basis vectors. For IP SAF, $\delta\mathbf{m_+} = \delta m_{+,\varphi}\mathbf{e_\varphi} + \delta m_{+,z}\mathbf{e_z}$, $\delta\mathbf{m_-} = \delta m_{-,\varphi}\mathbf{e_\varphi} + \delta m_{-,z}\mathbf{e_z}$, and for PMA SAF (T-Type), $\delta\mathbf{m_+} = \delta m_{+,x}\mathbf{e_x} + \delta m_{+,\theta}\mathbf{e_\theta}$, $\delta\mathbf{m_-} = \delta m_{-,x}\mathbf{e_x} + \delta m_{-,\theta}\mathbf{e_\theta}$. Then, the dynamic equation can be written in a uniform form:

$$i\frac{\omega}{\gamma}\begin{bmatrix} \delta m_{+,x(\varphi)} \\ \delta m_{+,\theta(z)} \\ \delta m_{-,x(\varphi)} \\ \delta m_{-,\theta(z)} \end{bmatrix} = \mathbf{A}_k \begin{bmatrix} \delta m_{+,x(\varphi)} \\ \delta m_{+,\theta(z)} \\ \delta m_{-,x(\varphi)} \\ \delta m_{-,\theta(z)} \end{bmatrix} = \begin{bmatrix} 0 & A_{12}^k & 0 & A_{14}^k \\ A_{21}^k & 0 & A_{23}^k & 0 \\ 0 & A_{32}^k & 0 & A_{34}^k \\ A_{41}^k & 0 & A_{43}^k & 0 \end{bmatrix} \begin{bmatrix} \delta m_{+,x(\varphi)} \\ \delta m_{+,\theta(z)} \\ \delta m_{-,x(\varphi)} \\ \delta m_{-,\theta(z)} \end{bmatrix}$$

$$(5)$$

where we use $k$ to label different calculation models. For $k = 1$ or 2, the system is considered based on the simplified case, where 1 corresponds to IP SAF, and 2 corresponds to PMA SAF (T-Type). And for $k = 3$ or 4, the system is considered based on the complete case, where 3 corresponds to IP SAF, and 4 corresponds to PMA SAF (T-Type). The detailed expressions of the matrix elements $A_{ij}^k$ are shown in Section S3. Resonance frequency $f_{res} = \omega_{res}/2\pi$ can then be solved from the secular equation of Eq. 5.

## Expressions of parameters in the generalized Hopfield model

The proof of Eq. 1 is shown in Section S5, where we show that the parameters in the generalized Hopfield model can be expressed in

terms of the matrix elements $A_{ij}^k$ in the classical method:

$$\omega_+ = \gamma\sqrt{-A_{12}^k A_{21}^k}$$

$$\omega_- = \gamma\sqrt{-A_{43}^k A_{34}^k}$$

$$g_1 = \frac{\gamma}{2}\sqrt{\frac{\gamma^2}{\omega_-\omega_+}\left[\frac{1}{4}(A_{32}^k A_{23}^k - A_{41}^k A_{14}^k)^2 + A_{21}^k A_{32}^k A_{43}^k A_{14}^k + A_{12}^k A_{23}^k A_{34}^k A_{41}^k\right] - A_{32}^k A_{23}^k - A_{41}^k A_{14}^k}$$

$$g_2 = \frac{\gamma}{2}\sqrt{\frac{\gamma^2}{\omega_-\omega_+}\left[\frac{1}{4}(A_{32}^k A_{23}^k - A_{41}^k A_{14}^k)^2 + A_{21}^k A_{32}^k A_{43}^k A_{14}^k + A_{12}^k A_{23}^k A_{34}^k A_{41}^k\right] + A_{32}^k A_{23}^k + A_{41}^k A_{14}^k}$$

$$(6)$$

Note that $g_2$ is not equal to $g_1$ in general unless $A_{32}^k A_{23}^k + A_{41}^k A_{14}^k = 0$. And $g_{1(2)}$ is not only related to the coupling terms in the matrix $\mathbf{A}_k$, but also related to pure + (−) mode terms $A_{12}^k$, $A_{21}^k$, $A_{34}^k$ and $A_{43}^k$. This is because the properties of + (−) mode magnons are determined by several parameters, including $H$ and $H_{ex}$, as discussed in Section S5.

Here, we consider the simplified case: $k = 1$ and 2. We point out that for IP SAF and PMA SAF, their normalized coupling strengths $g_1/\omega_0$ and $g_2/\omega_0$ have simple analytical expressions. In IP SAF, $g_1$ is equal to $g_2$ because of the zero $A_{23}^1$ and $A_{41}^1$. The expressions of $g_{1(2)}(H)$, $H_0$ and $\omega_0$ are shown below:

$$g_{1(2)}(H) = \frac{\gamma}{2}\left|\kappa\bar{H}_k\left[\left(1 - \frac{H^2}{4H_{ex}^2}\right)\frac{H^2}{\bar{H}_k(2H_{ex} + \bar{H}_k)}\right]^{1/4}\right|$$

$$H_0 = \sqrt{\frac{2H_{ex}^2\bar{H}_k}{H_{ex} + \bar{H}_k}} \quad (7)$$

$$\omega_0 = \gamma\sqrt{(2H_{ex} + \bar{H}_k)\frac{H_{ex}\bar{H}_k}{H_{ex} + \bar{H}_k}}$$

where $\kappa$ and $\bar{H}_k$ represent the degree of anisotropic asymmetry, $\kappa = (H_{k1} - H_{k2})/(H_{k1} + H_{k2})$, and the average anisotropy of the two FM, $\bar{H}_k = (H_{k1} + H_{k2})/2$, respectively. Therefore, the normalized coupling strengths of IP SAF can be expressed as

$$g_1/\omega_0 = g_2/\omega_0 = \frac{|\kappa|}{2}\sqrt{\left|\frac{\bar{H}_k}{\bar{H}_k + 2H_{ex}}\right|} \quad (8)$$

In PMA SAF, $H_0 = 0$ and $\omega_0 = (\gamma/2)\sqrt{(H_{k1} + H_{k2})(H_{k1} + H_{k2} - 4H_{ex})}$. When $H = 0$, $g_1$ and $g_2$ have analytical solutions: $g_1(H_0) = (\gamma/2)|H_{k1} - H_{k2}|$, $g_2(H_0) = 0$. Therefore, the normalized coupling strengths of PMA SAF can be expressed as

$$g_1/\omega_0 = \frac{|H_{k1} - H_{k2}|}{\sqrt{(H_{k1} + H_{k2})(H_{k1} + H_{k2} - 4H_{ex})}} = |\kappa|\sqrt{\left|\frac{\bar{H}_k}{\bar{H}_k - 2H_{ex}}\right|}$$

$$g_2/\omega_0 = 0 \quad (9)$$

For T-Type and the cases of $k = 3$ and 4, $g_1/\omega_0$ and $g_2/\omega_0$ are calculated numerically.

## Micromagnetic simulations

Micromagnetic simulations are carried out by using MuMax3[62]. The simulated dimension of the SAF structure is $400 \times 400 \times 6$ nm³ with periodic boundary condition applied along $x$ and $y$ direction. The used cell size is $4 \times 4 \times 3$ nm³ along the $x$, $y$, and $z$ direction. The magnetic parameters of the material are: saturation magnetization $M_S = 6 \times 10^5$ A/m, exchange stiffness constant $A_{ex} = 1.3 \times 10^{-11}$ J/m, and damping constant $\alpha = 0.01$. The two magnetic layers are characterized by various perpendicular magnetic anisotropy $K_{u1}$ and $K_{u2}$, respectively. And the RKKY coupling coefficient $J = -3.6 \times 10^{-4}$ J/m². The simulations are carried out in two steps: the static and dynamic steps. In the static simulation, the equilibrium magnetization configurations at each $H$ are obtained by minimizing the system's energy. In the

dynamic simulation, starting from the equilibrium magnetization configurations, a spatially uniform perturbation field is applied to excite the magnons. Here, a sinc type perturbation field is adopted as $\mathbf{h}_{\mathrm{rf}}(t) = (h_x\mathbf{e}_x + h_y\mathbf{e}_y)\sin(2\pi f_c t)/(2\pi f_c t)$ with the amplitude $h_x = h_y = 35$ Oe and the cutoff frequency $f_c = 100$ GHz. During the dynamic simulation of 10 ns, the spatially averaged magnetization $m(t)$ is recorded every 2 ps. Then the magnon spectrum is obtained by performing Fourier transform of $m(t)$.

## Sample structures and preparation process

The detailed multilayer film structures of our SAF samples are displayed below:

S1: Substrate//Ta (3)/Pt (3)/[Co (0.4)/Ni (0.8)]$_3$/Co (0.4)/Ir (0.64)/[Co (0.64)/Ni (1.28)]$_2$/Co (0.64)/Pt (3)/Ta (3);

S2: Substrate//Ta (3)/Pt (3)/ Ir (0.6)/[Co (0.4)/Ni (0.8)]$_3$/Co (0.4)/Ir (0.64)/Ni$_{80}$Fe$_{20}$ (3)/Pt (3)/Ta (3);

S3: Substrate//Ta (3)/Pt (3)/ Ir (0.6)/[Co (0.68)/Ni (1.36)]$_2$/Co (0.68)/Ir (0.64)/Ni$_{80}$Fe$_{20}$ (3)/Pt (3)/Ta (3);

S4: Substrate//Ta (3)/Pt (3)/ Ir (0.6)/ Co (3)/Ir (0.64)/Ni$_{80}$Fe$_{20}$ (3)/Pt (3)/Ta (3);

S5: Substrate//Ta (3)/Pt (3)/ Ir (0.6)/Co (1.36)/Ni (2.72)/Co (0.68)/Ir (0.64)/Ni$_{80}$Fe$_{20}$ (3)/Pt (3)/Ta (3);

S6: Substrate//Ta (3)/Pt (3)/[Co (0.4)/Ni (0.8)]$_3$/Co (0.4)/Ir (0.64)/[Co (0.4)/Ni (0.8)]$_3$/Co (0.4)/Pt (3)/Ta (3);

S7: Substrate//Ta (3)/Pt (3)/[Co (0.4)/Ni (0.8)]$_3$/Co (0.4)/Ir (0.64)/[Co (0.54)/Ni (1.08)]$_2$/Co (0.54)/Pt (3)/Ta (3), where the thicknesses in brackets are given in nanometer and the subscripts are the periods of Co/Ni stack. The aim of using 0.64 nm Ir between FM1 and FM2 is to provide a moderate RKKY interaction suitable for ST-FMR studies. The aim of inserting a 0.6 nm layer of Ir between Pt and Co in T-Type and IP SAF is to decrease the magnetic anisotropy of Co/Ni stack to a suitable value. By adjusting the components and the periods of Co/Ni stacks, the magnetic anisotropies of our SAF system can be greatly tuned.

To prepare the samples, we first deposit the films on thermally oxidized Si substrates by magnetron sputtering system (AJA) at room temperature. Then, the multilayers are patterned into $4 \times 18\,\mu\mathrm{m}^2$ rectangular strips by photolithography and Ar ion etching. Pt (10 nm)/Au (100 nm) ground-signal-ground (GSG) electrodes are then deposited by lift-off.

## ST-FMR measurements

The setup of ST-FMR measurement is shown in Fig. 3a. The samples are placed in a homemade electromagnet system which provides a magnetic field of up to 1 T. A modulated rf current $i_{\mathrm{STFMR}}$ is generated by a signal generator (R&S, SMB 100 A) and applied to the device. The modulation frequency is 231 Hz. Through the combined effects of spin-orbit torque and current-induced Oersted field, the magnetic moments begin to precess about the equilibrium axis. Then, due to the rectification effect, the device returns a modulated dc mixing voltage $V_{\mathrm{mix}}$ which is read out by lock-in amplifier (Stanford, SR850). A bias tee is used to separate the low frequency and rf signal from the mixed signal. The external magnetic field $H$ is applied in the film plane at an angle of 45° to the rf current direction in order to get a large $V_{\mathrm{mix}}$.

The dynamic responses of the samples can be obtained in two methods. The first method is to first fix the frequency $f$ of the rf signal, and record $V_{\mathrm{mix}}$ while sweeping the field $H$. Then change the frequency $f$ and repeat the above process. We refer to this method as the field-sweeping ST-FMR, which is commonly used to characterize the efficiency of spin torques[56]. Through fitting the resonance spectra, the resonance fields $H_r$ at different $f$ can be extracted.

In the other method, the field $H$ is first fixed, and $V_{\mathrm{mix}}$ is recorded while sweeping the frequency $f$ of rf signal. Then $H$ is changed, and the above process is repeated. We refer to this method as the frequency-

sweeping ST-FMR. By adopting this method, we can directly obtain the resonance frequencies of branch I and branch II at a certain $H$, such as $H = H_0$. Therefore, we adopt the frequency-sweeping ST-FMR in our experiments unless otherwise specified. When performing this method, a main problem is that the microwave power loss at high frequencies is larger than that at low frequencies, resulting in $V_{\mathrm{mix}}$ attenuation at high frequencies. To solve the problem, before measurements, we use the vector network analyzer to characterize the transmission loss $S_{21}(f)$ in our system. During the measurements, we adjust the output power of the signal generator according to $S_{21}(f)$ to compensate for the loss. In this way, the quality of the resonance spectra can be guaranteed even at high frequencies. In our experiments, the actual power applied to the devices is $-1.8$ dBm for samples S1 to S7. The Lorentzian functions are adopted to fit the experimental data. If only one branch appears in the spectra or if the two branches can be separately distinguished, we adopt an ordinary single Lorentzian function to extract $f_{\mathrm{res}}$. If the two branches are very close to each other in the spectra, we adopt the following multi-Lorentzian function to fit the experimental data:

$$V_{\mathrm{mix}} = V_{\mathrm{offset}} + V_S^h \frac{(\kappa_h/2\pi)^2}{(f - f_r^h)^2 + (\kappa_h/2\pi)^2} + V_A^h \frac{(\kappa_h/2\pi)(f - f_r^h)}{(f - f_r^h)^2 + (\kappa_h/2\pi)^2}$$
$$+ V_S^l \frac{(\kappa_l/2\pi)^2}{(f - f_r^l)^2 + (\kappa_l/2\pi)^2} + V_A^l \frac{(\kappa_l/2\pi)(f - f_r^l)}{(f - f_r^l)^2 + (\kappa_l/2\pi)^2} \quad (10)$$

where $f_r^{h(l)}$ is the resonance frequency of the branch I (II). $\kappa_{h(l)}/2\pi$ is the resonance frequency linewidth of the branch I (II). Figure 3b shows such a fitting process, where the resonance frequencies at a certain field can be extracted, as shown in the dashed lines.

## Data availability

All relevant data are available from the corresponding authors upon request.

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

## Acknowledgements

This work was supported by financial support from the National Key Research and Development Program of China (Grant No. 2022YFA1402800), the National Natural Science Foundation of China (Grants Nos. 12274437, 12074189, 52161160334), the Science Center of the National Science Foundation of China (Grant No. 52088101), the Beijing Natural Science Foundation (Grant No. Z190009), and the CAS Project for Young Scientists in Basic Research No. YSBR-084.

## Author contributions

Y.W. and G.Y. designed this study. F.M. and G.Y. supervised the study. Y.W. and W.Y. proposed the theory models. Y.W. and Z.Y. performed the analytical and numerical calculations. Y.Z., F.M., J.X., and Y.W. performed the micromagnetic simulations. Y.W., B.H., and W.H. prepared the experimental samples under the supervision of C.W. and X.H. J.W., Y.W., J.L., J.D., and H.X. built the ST-FMR equipment and provided the measurement programs. Y.W. and X.L. performed the ST-FMR & VSM measurements. Y.W. performed the VNA FMR measurements with the help of J.X. and H.X. C.L., and G.C. performed the BLS measurements. Y.W., Y.Z., C.L., G.Y., F.M. and P.Y. gave a hand in writing the manuscript. All the authors discussed the results and commented on the manuscript.

## Competing interests

The authors declare no competing interests.
