## [Peer Review File · Nature Communications]

Ultrastrong to nearly deep-strong magnon-magnon coupling with a high degree of freedom in synthetic antiferromagnetsREVIEWER COMMENTS

Reviewer #1 (Remarks to the Author):

The paper by Wang et al. entitled 'Ultrastrong to nearly deep-strong magnon-magnon coupling with a high degree of freedom in synthetic antiferromagnets' investigated the room-temperature magnon-magnon ultrastrong coupling in synthetic antiferromagnets (SAF). They theoretically and experimentally estimated the coupling strength g_1 and g_2 in in-plane, perpendicular, T-shape type SAF configuration. This large coupling strength and g/ω_0 are very interesting but are not entirely unexpected in the MFM1+MFM2 (+ mode) and MFM1-MFM2 (- mode) case. The reviewer feels that this manuscript is suitable for publication in Nature Communications after the following points are clarified.

It is usually to choose the sublattice spins or combined magnetic moments with certain physics meaning. For example, counterclockwise and clockwise precession are identified as spin-up and spin-down magnons (PHYSICAL REVIEW LETTERS 123, 2019, 117204), and in-phase (out-of-phase) precession magnetic moment are identified as acoustic mode (optical mode) (Adv. Funct. Mater. 2023, 2303781). In this manuscript, the author discusses the coupling of $M_+=MFM1+MFM2$ (+ mode) and $M_-=MFM1-MFM2$ (- mode) magnons. The physics meaning of these modes should be further explained especially in the T-shape magnetization type.

From both the classical marcospin approach and quantum description of SAF system, the coupling of + mode and - mode is mainly origin from the Zeeman energy and anisotropy energy which are determined by magnet moment of each FM layer. While, if ignore quantities above second order, the exchange interaction term can't appear in the coupling term (H_{couple}), because the M_+ M_- just includes the terms of MFM12 and MFM22. Is it the proper choice of the + mode and - mode to estimate the magnon-magnon coupling in a system by only considering the contribution of Zeeman effect and anisotropic energy, and is it possible to choose other linear combination of MFM1 and MFM2 in the system, such as $MFM1-2MFM2$ and $MFM1+2MFM2$, to achieve a higher coupling strength value (g)?

I have some other minor concerns about the manuscript:

In the classical marcospin approach, there is exchange interaction term in the g_1 and g_2 (equation 7 and equation 9, page 17), while in the quantum model, the exchange interaction is not included (equation S-8, supplementary page 8). Is there any difference between g_1 and g_2 calculated by these two methods?

In the Figure 4a-c, the author claimed that they fitted the $\omega_{\perp I}$ and $\omega_{\perp II}$ with H , and the Hex1 and Hex2 are the fitting parameters. While the reviewer could only find the analytical expression (equation S-20 and equation 7-9) for Figure 4a and Figure 4b. Moreover, the Hex2 is not included in the expression.

In line 732 and 733, page 29, the author claimed, in Figure a-c, the g_1/ω_0 are determined to be 0.393, 0.972, 0.278, and g_2/ω_0 are determined to be 0.489, 0, 0.706, respectively. While, from the reviewer's understanding, in Figure a-c, g/ω_0 is also dependent on the H .

The author should compare the calculation results of g/ω_0 from ST-FMR fitting parameters and hysteresis loops parameters.

The reviewer found there many intermediate states of the SAF from the VSM measurement (Figure. S9, page 34 in supplementary). Dose the magnetic moment keep single domain during ST-FMR measurement especially when the external magnetic field is zero?

Reviewer #2 (Remarks to the Author):

This paper investigates the coupling of antiferromagnetic magnon modes in synthetic antiferromagnets with various magnetization configuration types by tuning the perpendicular magnetic anisotropy, both theoretically and experimentally. In particular, the theoretical discussion of magnon-magnon coupling from both classical and quantum perspectives is novel and has rarely been reported so far. Because the magnon-based hybrid systems, including the magnon-magnon coupling, have recently been received much attention due to its unique functionality, this study may provide an opportunity to take a new insight into that field from a quantum perspective.

However, regarding the experimental part, although the ST-FMR measurements in SAFs have been measured, the resonance peaks are unclear from the color plots of the ST-FMR spectra and it is difficult to say that the theoretical considerations are substantiated. In addition, the estimation of the coupling strength from the experimental results is unclear in many respects. Therefore, I am not sure that this work is of sufficient interest and significance to reach high criteria for publications in Nature Communications.

Please consider the several concerns as mentioned below.

1. Line 6 on Page 7, the authors state “By tuning the anisotropies, the gap can reach the maximum as the magnetization configuration is almost converted from IP SAF to T-type, as shown in Fig. 1d”, but Fig. 1(d) shows the resonance frequencies or extracted coupling strength as a function of magnetic field, not tuning the magnetic anisotropy. In addition, the magnetization configuration in IP-SAF should change from a canted configuration to a parallel configuration as increasing the magnetic field, and it does not convert to a T-type SAF. Therefore, the above statement is not consistent with the results shown in Fig. 1(d).
2. Regarding the T-type SAFs shown in Fig. 1(f), $H = 0$ [Oe] was defined as the magnetic field strength at the center frequency, $H_{\{0\}}$. But the coupling strengths of the magnon-magnon coupling in T-type SAF should not be defined, because the resonance frequencies of decoupled – and + modes are not crossed, as shown in dashed lines in Fig. 1(f). The coupling strength obtained at such a magnetic field strength are completely meaningless.
3. Is there any effect of the magnetization configuration in PMA and T-type SAFs? For example, in the case of PMA SAF, the magnetic resonance precession directions of low and high frequency modes are reversed in the head-to-head magnetization configuration and tail-to-tail magnetization configuration, as discussed in the previous paper (e.g. Y. Shiota et al., Phys. Rev. Applied 18, 014032 (2022)).
4. The ST-FMR spectrum obtained in the experiments shown in Fig. 3(b) does not indicate which sample was measured.
5. The spectrum fitting using the multi-Lorentzian functions have been performed. But there is no discussion about the magnon dissipation, excitation efficiency etc. And no spectrum fittings have been done for the other ST-FMR spectra.
6. Regarding the experimental results of the color plots in Fig. 4, the resonance peaks (especially the resonance of the high-frequency mode) are unclear, and it is difficult to determine whether they are consistent with the theoretical calculations. It is written as if the coupling strength is obtained experimentally, but in reality, it seems that the coupling strength is only obtained by theoretical or numerical calculations using material parameters estimated from M-H curves and other data.

We extend our sincere appreciation to both reviewers for their recognition of our work and their provision of invaluable insights through their comments. We have made significant improvements in both the theory and experiments. Especially in the experimental section, we have improved our experimental methods and re-measured all the samples. Besides, we also adopt new experimental methods as a supplement. We have integrated our responses into the revised version of the manuscript, where we use the blue color to highlight the changes related to the comments. We hope these responses can effectively address the reviewers' concerns. Please see below for our detailed point-by-point response to the reviewers' comments.

Response to Reviewer #1

Comment: The paper by Wang et al. entitled 'Ultrastrong to nearly deep-strong magnon-magnon coupling with a high degree of freedom in synthetic antiferromagnets' investigated the room-temperature magnon-magnon ultrastrong coupling in synthetic antiferromagnets (SAF). They theoretically and experimentally estimated the coupling strength g_1 and g_2 in in-plane, perpendicular, T-shape type SAF configuration. This large coupling strength and g/ω_0 are very interesting but are not entirely unexpected in the $M_{FM1}+M_{FM2}$ (+ mode) and $M_{FM1}-M_{FM2}$ (- mode) case. The reviewer feels that this manuscript is suitable for publication in Nature Communications after the following points are clarified.

Response: We thank the reviewer for recognizing our work and recommending its publication after clarifying the points.

Comment 1: *It is usually to choose the sublattice spins or combined magnetic moments with certain physics meaning. For example, counterclockwise and clockwise precession are identified as spin-up and spin-down magnons (PHYSICAL REVIEW LETTERS 123. 2019, 117204), and in-phase (out-of-phase) precession magnetic moment are identified as acoustic mode (optical mode) (Adv. Funct. Mater. 2023, 2303781). In this manuscript, the author discusses the coupling of $M^+=M_{FM1}+M_{FM2}$ (+ mode) and M^-*

= $M_{FM1}-M_{FM2}$ (- mode) magnons. The physics meaning of these modes should be further explained especially in the T-shape magnetization type.

Response 1: Thanks for the reviewer’s comment. Actually, the + mode and – mode in our work are **optical** and **acoustic** modes, respectively. We need to point out that both M_i^+ and M_i^- are not the *direct* addition and subtraction of the components of \mathbf{M}_{S1} and \mathbf{M}_{S2} . As shown in “Section S5: *Quantum description of SAF system*” in the revised Supplementary Materials, we define an operator C_{2r} which rotates vectors 180° about the r axis, where the r axis is defined as $\mathbf{e}_r \parallel \mathbf{m}_1^{eq} + \mathbf{m}_2^{eq}$, and then we apply this operator C_{2r} to \mathbf{M}_{S2} , which means we rotate \mathbf{M}_{S2} 180° around the r axis. Then we get a new vector: $\mathbf{M}'_{S2} = C_{2r}\mathbf{M}_{S2}$. M_i^+ (M_i^-) is defined as the addition (subtraction) of the components of \mathbf{M}_{S1} and $\mathbf{M}'_{S2} = C_{2r}\mathbf{M}_{S2}$. Under this kind of definition, M_i^+ and M_i^- directly correspond to the macrospin joint precession vectors $\delta\mathbf{m}_+ = \delta\mathbf{m}_1 + C_{2y}\delta\mathbf{m}_2$ and $\delta\mathbf{m}_- = \delta\mathbf{m}_1 - C_{2y}\delta\mathbf{m}_2$ as shown in *Materials and Methods* part of the main text, respectively. While the two macrospin vectors $\delta\mathbf{m}_+$ and $\delta\mathbf{m}_-$ are invariant (odd) under the combination of twofold rotation C_{2y} and sublattice exchange, corresponding to pure optical and acoustic modes, respectively. Therefore, the + and – modes in our work correspond to optical and acoustic modes, respectively. The adapted definition applies to all three magnetization configurations in our work: PMA SAF, IP SAF, and T-Type.

Actually, in our work, we believe that the Hamiltonian expressed in Eq. 1 is universally applicable to the coupling between two different magnon modes and is not only limited to the coupling between optical and acoustic magnon modes. Therefore, we use the two abstract symbols “+” and “–” to represent the two pure modes involved in the coupling, which can have different meanings in different hybrid systems. And we hope that such a quantum description can be applied to other magnonic systems.

To avoid misunderstandings, we have added the following Fig. R1 to Fig. 1d as inset in the revised manuscript and added suggested descriptions, *i.e.*, “**out-of-phase**” and “**in-phase**”, on Page 6, Lines 142-143.

We also modified the “Section S5: *Quantum description of SAF system*” in the

revised Supplementary Materials.

Figure R1. Schematics of the + (optical) mode and the – (acoustic) mode.

Comment 2: *From both the classical macrospin approach and quantum description of SAF system, the coupling of + mode and – mode is mainly origin from the Zeeman energy and anisotropy energy which are determined by magnet moment of each FM layer. While, if ignore quantities above second order, the exchange interaction term can't appear in the coupling term (\mathcal{H}_{couple}), because the $M^+ M^-$ just includes the terms of M_{FM1}^2 and M_{FM2}^2 . Is it the proper choice of the + mode and – mode to estimate the magnon-magnon coupling in a system by only considering the contribution of Zeeman effect and anisotropic energy, and is it possible to choose other linear combination of M_{FM1} and M_{FM2} in the system, such as $M_{FM1}-2M_{FM2}$ and $M_{FM1}+2M_{FM2}$, to achieve a higher coupling strength value (g)?*

Response 2: Thanks for the reviewer's comment. Since the + mode and – mode refer to the optical and acoustic modes in our work as mentioned in the above response, it is known that the optical and acoustic modes are combined modes of FM1 and FM2, while the RKKY exchange interaction acts between FM1 and FM2. Therefore, the RKKY exchange interaction term mainly influences the properties of pure + and – modes, and does not directly influence the coupling between the + mode and – mode, as shown in Fig. R2a. Thus $\mathcal{H}_{Exchange}$ term does not necessarily have to appear in \mathcal{H}_{Couple} term.

We consider a previously studied case to further illustrate this point. In some previous works, an oblique magnetic field \mathbf{H} is applied to the sample to break the rotational symmetry, thus leading to the coupling between the optical (+) and acoustic (–) modes [see: Physical Review Letters 123, 047204 (2019); Physical Review B 102,

100403 (2020)]. For instance, we consider an intrinsic symmetric IP SAF, and apply such an oblique \mathbf{H} to the SAF, as shown in Figs. R2b and R2c. Then, we apply a similar quantum description to this system. The classical Hamiltonian of this system can be written as

Figure R2. **a** Schematic of the coupling between the + mode and - mode. **b** and **c** The coordinate systems for a previously studied case where an oblique magnetic field \mathbf{H} is applied to induce the magnon-magnon coupling. **(b)** corresponds to the spatial configuration of the system, where θ corresponds to the angle between the $+z$ axis and \mathbf{H} , θ' corresponds to the angle between the $+z$ axis and r axis, and φ corresponds to the angle between the $+x$ axis and \mathbf{m}_2^{eq} . The blue dashed line corresponds to the projection of \mathbf{m}_2^{eq} on the x - O - y plane. **(c)** corresponds to the x - O - r plane. Two local coordinate systems are defined, which are consistent with Fig. S5 in the revised Supplementary Materials.

$$\mathcal{H} = \mathcal{H}_{\text{Zeeman}} + \mathcal{H}_{\text{Ani}} + \mathcal{H}_{\text{Exchange}} \quad (\text{R1})$$

with

$$\mathcal{H}_{\text{Zeeman}} = -\mu_0 H \left[\sin(\theta' - \theta) M_{x'}^- + \cos(\theta' - \theta) \cos \varphi M_{y'}^+ + \cos(\theta' - \theta) \sin \varphi M_{z'}^+ \right]$$

$$\begin{aligned}
\mathcal{H}_{\text{Ani}} &= -\mu_0 \frac{H_k}{4M_S} \left[\sin^2 \theta' (M_{x'}^{+2} + M_{x'}^{-2}) + \cos^2 \varphi \cos^2 \theta' (M_{y'}^{+2} + M_{y'}^{-2}) \right. \\
&\quad + \sin^2 \varphi \cos^2 \theta' (M_{z'}^{+2} + M_{z'}^{-2}) + \cos \varphi \sin 2\theta' (M_{x'}^+ M_{y'}^- + M_{x'}^- M_{y'}^+) \\
&\quad \left. + \sin \varphi \sin 2\theta' (M_{x'}^+ M_{z'}^- + M_{x'}^- M_{z'}^+) + \sin 2\varphi \cos^2 \theta' (M_{y'}^+ M_{z'}^+ + M_{y'}^- M_{z'}^-) \right] \\
\mathcal{H}_{\text{Exchange}} &= -\mu_0 \frac{H_{\text{ex}}}{4M_S} \left[-M_{x'}^{+2} + M_{x'}^{-2} + \cos 2\varphi (M_{y'}^{+2} - M_{y'}^{-2} - M_{z'}^{+2} + M_{z'}^{-2}) \right. \\
&\quad \left. + 2\sin 2\varphi (M_{y'}^+ M_{z'}^+ - M_{y'}^- M_{z'}^-) \right]
\end{aligned}$$

which can be divided into two parts:

$$\mathcal{H} = \mathcal{H}_{\text{Pure}} + \mathcal{H}_{\text{Couple}} \quad (\text{R2})$$

with

$$\begin{aligned}
\mathcal{H}_{\text{Pure}} &= -\mu_0 H \left[\sin(\theta' - \theta) M_{x'}^- + \cos(\theta' - \theta) \cos \varphi M_{y'}^+ + \cos(\theta' - \theta) \sin \varphi M_{z'}^+ \right] \\
&\quad - \mu_0 \frac{H_k}{4M_S} \left[\sin^2 \theta' (M_{x'}^{+2} + M_{x'}^{-2}) + \cos^2 \varphi \cos^2 \theta' (M_{y'}^{+2} + M_{y'}^{-2}) \right. \\
&\quad \left. + \sin^2 \varphi \cos^2 \theta' M_{z'}^{+2} + 2 \sin \varphi \sin 2\theta' M_S M_{x'}^- + 2 \sin 2\varphi \cos^2 \theta' M_S M_{y'}^+ \right] \\
&\quad - \mu_0 \frac{H_{\text{ex}}}{4M_S} \left[-M_{x'}^{+2} + M_{x'}^{-2} + \cos 2\varphi (M_{y'}^{+2} - M_{y'}^{-2} - M_{z'}^{+2}) + 4\sin 2\varphi M_S M_{y'}^+ \right] \\
\mathcal{H}_{\text{Couple}} &= -\mu_0 \frac{H_k}{4M_S} \cos \varphi \sin 2\theta' (M_{x'}^+ M_{y'}^- + M_{x'}^- M_{y'}^+)
\end{aligned}$$

From Eqs. R1 and R2, it can be seen that $\mathcal{H}_{\text{Exchange}}$ also does not appear in $\mathcal{H}_{\text{Couple}}$ in this previously studied system. We point out that the main function of $\mathcal{H}_{\text{Exchange}}$ is to construct the pure + mode and - mode, thus affecting the center frequency ω_0 . In Section S5: *Quantum description of SAF system* in the revised Supplementary Materials, besides the quantum description of PMA SAF, we have also added the quantum description of our IP SAF system for completeness. In this case, the $\mathcal{H}_{\text{Couple}}$ can be expressed as

$$\mathcal{H}_{\text{Couple}} = -\frac{\mu_0}{8M_S} (H_{k1} - H_{k2}) (M_{x'}^+ M_{x'}^- + M_{x'}^- M_{x'}^+) \quad (\text{R3})$$

We find that $\mathcal{H}_{\text{Couple}}$ in our asymmetric IP SAF case also has no component of $\mathcal{H}_{\text{Exchange}}$ and includes only the anisotropy energy \mathcal{H}_{Ani} . All these examples show that

it is common and reasonable that $\mathcal{H}_{\text{Exchange}}$ does not appear in $\mathcal{H}_{\text{Couple}}$ when considering the coupling between the optical and acoustic modes.

However, we must point out that although $\mathcal{H}_{\text{Exchange}}$ term does not appear in $\mathcal{H}_{\text{Couple}}$ term, the $\mathcal{H}_{\text{Exchange}}$ term can influence the values of g_1 and g_2 . Therefore, the expected g_1 and g_2 will change if we do not consider the RKKY interaction energy in a system. As this is also be concerned in Comment 3, we will explain this in detail in the next response.

Regarding whether it is possible to achieve a higher g , we consider the case where the $+$ mode and $-$ mode intersect. When $H = H_0$, the center frequency $\omega_0 = \omega_+ = \omega_-$. By substituting ω_0 into Eq. S-23 in the revised Supplementary Materials, we obtain:

$$\omega_{\text{II}}^2 = (\omega_0 \pm g_1)^2 - g_2^2 \quad (\text{R4})$$

When $\omega_{\text{II}} = 0$, g_1 and g_2 have the following relationship:

$$g_2 = |\omega_0 - g_1| \quad (\text{R5})$$

which can be divided into two cases: for the case $g_2 = \omega_0 - g_1$, we can get:

$$\frac{g_1}{\omega_0} + \frac{g_2}{\omega_0} = 1 \quad (\text{R6})$$

While for the other case $g_2 = g_1 - \omega_0$, we can get:

$$\frac{g_1}{\omega_0} - \frac{g_2}{\omega_0} = 1 \quad (\text{R7})$$

Equations R5 and R6 indicate the maximum $g_{1(2)}/\omega_0$ that can be realized. We point out that our system belongs to the first case, and we have achieved the maximum $g_{1(2)}/\omega_0$, *i.e.*, $g_1/\omega_0 = 1$. However, for the second case, we note that the maximum g_1/ω_0 can be greater than 1. For magnon-magnon hybrid systems, whether the second case can be achieved remains to be explored.

We further consider the modes proposed by the reviewer: $M_{i'}^{+'} = M_{S1,i'} + 2M_{S2,i'}$ and $M_{i'}^{-'} = M_{S1,i'} - 2M_{S2,i'}$ in IP SAF systems. The resonance frequencies of the pure $+$

mode and $-$ mode, in this case, can be written as

$$\begin{aligned}\omega_{+,\cdot} &= \gamma \sqrt{-\left(-\frac{9}{4}H_{\text{ex}} + \frac{5H^2}{8H_{\text{ex}}}\right)\left(-\frac{1}{4}H_{\text{ex}} + \frac{H_{k1} + H_{k2}}{2}\right)} \\ \omega_{-,\cdot} &= \gamma \sqrt{-\left(\frac{1}{4}H_{\text{ex}} - \frac{5H^2}{8H_{\text{ex}}}\right)\left(\frac{9}{4}H_{\text{ex}} + \frac{H_{k1} + H_{k2}}{2}\right)}\end{aligned}\quad (\text{R8})$$

From Eq. R8, we find that the value of $\omega_{-,\cdot}$ is imaginary when H is small. This imaginary $\omega_{-,\cdot}$ implies that this pair of modes do not actually exist. Nevertheless, the reviewer provides very meaningful insights. For other linear combinations, they still have the possibility of corresponding to actually existed modes. This possibility and the coupling between them remain to be further investigated.

***Comment 3:** In the classical macrospin approach, there is exchange interaction term in the g_1 and g_2 (equation 7 and equation 9, page 17), while in the quantum model, the exchange interaction is not included (equation S-8, supplementary page 8). Is there any difference between g_1 and g_2 calculated by these two methods?*

Response 3: Thanks for the reviewer's comment. We point out that Eqs. 6 to 9 are actually derived from the quantum model, as g_1 and g_2 are both concepts derived from the quantum model. While the derived g_1 and g_2 can be expressed in terms of the matrix elements A_{ij}^k , as mentioned in Eq. 6 in the revised manuscript. Therefore, by using Eq. 6, the quantum model can be linked to the parameters in the macrospin matrix. From Eq. 6, we note that $g_{1(2)}$ is not only linked to the coupling terms in the matrix \mathbf{A}_k but also linked to pure $+$ ($-$) mode terms A_{12}^k , A_{21}^k , A_{34}^k , and A_{43}^k . Due to the presence of H_{ex} in the pure mode terms, $g_{1(2)}$ is related to H_{ex} . We continue to explain the reasons behind the correlation between $g_{1(2)}$ and H_{ex} . As we mentioned in response to comment 2, although $\mathcal{H}_{\text{Exchange}}$ term does not appear in $\mathcal{H}_{\text{Couple}}$ term, this does not mean that g_1 and g_2 are independent of the parameter H_{ex} . We point out two reasons for this result. The minor reason is that $\mathcal{H}_{\text{Couple}}$ for PMA SAF (T-Type) includes θ_1 and θ_2 , while $\theta_{1(2)}$ is solved numerically from Eq. S-6, which is a function of H_{ex} and other parameters.

Thus, this reason partly accounts for the result that the coupling strengths are related to H_{ex} in the PMA SAF (T-Type) system. And the major reason is that in Heisenberg equations of motion Eq. S-17, the matrix elements A_{pure}^k , B_{pure}^k , C_{pure}^k , D_{pure}^k are not constants but functions of the parameters of the system, including H_{ex} and the external field. Thus, the solved Bogoliubov transformation coefficients $w_{+(-)}^k$, $x_{+(-)}^k$, $y_{+(-)}^k$, $z_{+(-)}^k$ are also not constants but functions of the parameters of the system and the external field. Therefore, when applying the inverse transformation Eq. S-16 to $\hat{\mathcal{H}}_{\text{Couple}}$, H_{ex} is introduced into the expression of $g_{1(2)}$.

We have added the following sentences on Page 18, Lines 471 to 474 in the revised manuscript as:

“And $g_{1(2)}$ is not only related to the coupling terms in the matrix \mathbf{A}_k , but also related to pure + (–) mode terms A_{12}^k , A_{21}^k , A_{34}^k and A_{43}^k . This is because the properties of + (–) mode magnons are determined by several parameters, including H and H_{ex} , as discussed in Section S5.”

We have also added the corresponding analysis to Section S5 in the revised Supplementary Materials.

In addition, although $g_{1(2)}$ is related to H_{ex} , we note that H_{ex} only plays a minor role in changing the values of g_1/ω_0 and g_2/ω_0 compared to the magnetic anisotropies. Detailed analysis can be found in updated Section S6 in the revised Supplementary Materials, and we have added the sentences on Page 10, Lines 256-262 in the revised manuscript as:

“We also investigate the effect of RKKY interaction strength on g_1/ω_0 and g_2/ω_0 , as shown in Section S6. The results show that most of the properties mentioned above hold true when H_{ex} changes, and H_{ex} only plays a minor role in changing the value of g_1/ω_0 and g_2/ω_0 . This is because the pure + mode and – mode are combined modes of FM1 and FM2, while the RKKY interaction acts between FM1 and FM2. Therefore, the RKKY interaction does not directly influence the coupling between the + mode and

– mode.”

Comment 4: *In the Figure 4a-c, the author claimed that they fitted the ω_l and ω_{ll} with H , and the H_{ex1} and H_{ex2} are the fitting parameters. While the reviewer could only find the analytical expression (equation S-20 and equation 7-9) for Figure 4a and Figure 4b. Moreover, the H_{ex2} is not included in the expression.*

Response 4: We thank the reviewer for pointing out this point. The expressions for ω_l , ω_{ll} , the pure + and – modes, and $g_{1(2)}(H)$ in Fig. 4a to 4c are shown in **Eq. 6, Section S3**, and **Eq. S-23** in the revised manuscript and Supplementary Materials. For clarity, we provide the following explanation: As shown in Section S3, we display the specific expressions of matrix elements A_{ij}^k , where we use k to label different calculation models. For $k = 1$ or 2 , the system is considered based on the simplified case, where 1 corresponds to IP SAF, and 2 corresponds to PMA SAF (T-Type). And for $k = 3$ or 4 , the system is considered based on the complete case, where 3 corresponds to IP SAF, and 4 corresponds to PMA SAF (T-Type). Therefore, Figure 4a corresponds to the case of $k = 3$, and Figures 4b and 4c correspond to the case of $k = 4$. And the angles $\varphi_{1(2)}$ and $\theta_{1(2)}$ are derived from Eq. S-7 and Eq. S-8, which need to be solved numerically. As for the Eqs. 7-9, they represent the analytical coupling strengths and normalized coupling strengths in the simplified case as shown in theoretical section. Thus, the H_{ex2} is not included.

To avoid misunderstandings, we have added “The fittings are based on the complete case, where” and “as shown in Eq. 6, Section S3, and Eq. S-23” in Section S12: *Detailed fitting process and comparison of resonance spectrum fitting results with VSM fitting results* in the revised Supplementary Materials.

The following sentences are a supplement for the convenience of the reviewer: in our calculations, two cases are considered. One case is the simplified case, where we assume FM1 and FM2 share the same saturation magnetization M_S and thickness d and set J_2 to be zero. In other words, we only consider the effects caused by the asymmetry of magnetic anisotropy alone. The other case is the complete case, where the asymmetry

of magnetic moments of FM1 and FM2, as well as the biquadratic exchange interaction between FM1 and FM2, is considered. In the theoretical section, we mainly consider the simplified case in order to capture the essence of coupling caused by pure magnetic anisotropic asymmetry. While in the experimental section, we adopt the complete case to fit the experimental data to match the actual situation. We have mentioned in the manuscript that most of the coupling properties proposed in the simplified case remain unchanged for the complete case. In Section S11 in the revised Supplementary Materials, we show that the asymmetry of magnetic moments of FM1 and FM2 $\kappa' = (M_{s1}d_1 - M_{s2}d_2)/(M_{s1}d_1 + M_{s2}d_2)$, and the biquadratic exchange interaction only play a minor role in inducing USC in our samples compared to the asymmetry of magnetic anisotropy.

We have added the following sentences on Page 6, Lines 145-147, in the revised manuscript as:

“In this theoretical section, we show the simplified case, in which we only consider the effects caused by the asymmetry of magnetic anisotropy alone, as detailed in Materials and Methods.”

We have also added the following sentences on Page 12, Lines 302-303 in the revised manuscript as:

“the complete case is adopted in the fitting,”

Comment 5: *In line 732 and 733, page 29, the author claimed, in Figure a-c, the g_1/ω_0 are determined to be 0.393, 0.972, 0.278, and g_2/ω_0 are determined to be 0.489, 0, 0.706, respectively. While, from the reviewer's understanding, in Figure a-c, g/ω_0 is also dependent on the H .*

Response 5: Thanks for the reviewer's comment. For the coupling strengths g_1 and g_2 , they are indeed related to H , as shown in Figs. 1d to 1f and Figs. 4d to 4f in the revised manuscript. In our work, we mainly focus on the coupling properties at the coupling field, *i.e.*, magnetic field H_0 corresponding to the center frequency ω_0 . And for the normalized coupling strengths g_1/ω_0 and g_2/ω_0 , they are actually the abbreviations of

“ $g_1(H = H_0)/\omega_0$ ” and “ $g_2(H = H_0)/\omega_0$ ”, respectively. Therefore, the $g_{1(2)}$ in $g_{1(2)}/\omega_0$ actually refers to the $g_{1(2)}$ at $H = H_0$.

To make this point clear, we have added “here, we mainly focus on the coupling properties at \$H = H_0\$.” and “For convenience, we use \$g_{1,2}/\omega_0\$ to refer to \$g_{1,2}(H_0)/\omega_0\$ in the following.” on Page 9, Lines 222-225 in the revised manuscript.

Comment 6: *The author should compare the calculation results of g/ω_0 from ST-FMR fitting parameters and hysteresis loops parameters.*

Response 6: Thanks for the reviewer’s comment. From ST-FMR fitting parameters, for samples S1 to S3, g_1/ω_0 are determined to be 0.963, 0.232, 0.392, and g_2/ω_0 are determined to be 0, 0.597, 0.487, respectively. From hysteresis loops parameters, for samples S1 to S3, g_1/ω_0 are determined to be 0.987, 0.237, 0.402, and g_2/ω_0 are determined to be 0, 0.602, 0.484, respectively. We note that g_1/ω_0 and g_2/ω_0 extracted from the resonance spectra are consistent with g_1/ω_0 and g_2/ω_0 calculated from the parameters obtained through the hysteresis loops, with only very small differences, indicating the consistency of various methods in extracting parameters. According to the comments from Reviewer #2, we have re-prepared, re-measured, and re-fitted all the samples and also changed the definition of ω_0 in the T-Type case, the $g_{1(2)}/\omega_0$ extracted from ST-FMR here are different from that in the original manuscript. However, for IP SAF and PMA SAF cases, there are only very small differences in g_1/ω_0 and g_2/ω_0 between the revised and original versions, indicating the stability of our results.

We have added the following sentences on Page 13, Lines 324-327 in the revised manuscript as:

“We note that \$g_1/\omega_0\$ and \$g_2/\omega_0\$ extracted from the spectra here are consistent with \$g_1/\omega_0\$ and \$g_2/\omega_0\$ calculated from the parameters obtained through the hysteresis loops, as shown in Table S1, indicating the consistency of various methods in extracting parameters.”

We have also shown the $g_{1(2)}/\omega_0$ from ST-FMR fitting parameters and hysteresis

loop parameters in Table S1 in the revised Supplementary Materials.

Comment 7: *The reviewer found there are many intermediate states of the SAF from the VSM measurement (Figure. S9, page 34 in supplementary). Dose the magnetic moment keep single domain during ST-FMR measurement especially when the external magnetic field is zero?*

Response 7: Thanks for the reviewer's comment. We use Kerr microscopy to see the domain structures in continuous thin films, as shown in Fig. R3, where Figs. R3a to R3f correspond to PMA SAF S1 and Figs. R3g to R3l correspond to T-Type S2. The magnetic field H_{OOP} is applied perpendicular to the sample plane. Figures R3a and R3g show the hysteresis loops measured by Kerr microscopy, which correspond to VSM measured Figs. S15d and S15e, respectively. We take the images while sweeping the magnetic field from negative to positive, as shown in Figs. R3b to R3f and Figs. Rh to Rl. For the PMA SAF sample, we note that the single domain feature is kept in the image area when H_{OOP} sweeps from negative to 0 Oe and then to about 700 Oe. For the T-Type sample, the single domain feature is kept in the image area when H_{OOP} sweeps from negative to 0 Oe. Therefore, the results indicate the single domain feature for PMA SAF and T-Type when the external magnetic field is zero. We admit that due to the equipment limitations, we cannot apply an in-plane field, like the ST-FMR measurement, to see the domain structures. However, our re-measured ST-FMR results show that good spectra can still be obtained when the magnetic field is almost zero, as shown in Figs. S17b and S17c in the revised Supplementary Materials. This result implies that the single domain feature is also preserved in our small ST-FMR devices ($4 \times 10 \mu\text{m}^2$).

Figure R3. **a** to **f** Result of PMA SAF, where **(a)** corresponds to the hysteresis loops measured by Kerr microscopy and **(b)** to **(f)** correspond to the Kerr microscope images at different H_{OOP} . **g** to **l** Result of T-Type, where **(g)** corresponds to the hysteresis loops measured by Kerr microscopy and **(h)** to **(l)** correspond to the Kerr microscope images at different H_{OOP} .

Response to Reviewer #2

This paper investigates the coupling of antiferromagnetic magnon modes in synthetic antiferromagnets with various magnetization configuration types by tuning the perpendicular magnetic anisotropy, both theoretically and experimentally. In particular, the theoretical discussion of magnon-magnon coupling from both classical and quantum perspectives is novel and has rarely been reported so far. Because the magnon-based hybrid systems, including the magnon-magnon coupling, have recently been received much attention due to its unique functionality, this study may provide an opportunity to take a new insight into that field from a quantum perspective.

However, regarding the experimental part, although the ST-FMR measurements in SAFs have been measured, the resonance peaks are unclear from the color plots of the ST-FMR spectra and it is difficult to say that the theoretical considerations are substantiated. In addition, the estimation of the coupling strength from the experimental results is unclear in many respects. Therefore, I am not sure that this work is of sufficient interest and significance to reach high criteria for publications in Nature Communications.

Please consider the several concerns as mentioned below.

Response: We thank the reviewer for acknowledging that the theoretical discussion of magnon-magnon coupling from both classical and quantum perspectives is novel in our work, and our work may provide an opportunity to take a new insight into the magnon-related coupling from a quantum perspective. Regarding the experimental part, we appreciate the helpful insights provided by the reviewer. In the revised manuscript, we have made significant improvements to our ST-FMR measurements, and we have re-measured all the samples to obtain convincing results. Besides, we have also performed the Brillouin light scattering (BLS) measurements as a supplement to the ST-FMR measurements. Our point-to-point responses are given as follows.

Comment 1: *Line 6 on Page 7, the authors state “By tuning the anisotropies, the gap can reach the maximum as the magnetization configuration is almost converted from*

IP SAF to T-type, as shown in Fig. 1d”, but Fig. 1(d) shows the resonance frequencies or extracted coupling strength as a function of magnetic field, not tuning the magnetic anisotropy. In addition, the magnetization configuration in IP-SAF should change from a canted configuration to a parallel configuration as increasing the magnetic field, and it does not convert to a T-type SAF. Therefore, the above statement is not consistent with the results shown in Fig. 1(d).

Response 1: We thank the reviewer for pointing out this issue. We agree with the reviewer that our description may mislead the readers. We have revised the related statements as:

“By selecting the appropriate values of magnetic anisotropies, the IP SAF configuration can be adjusted to a critical case, for instance, $H_{k1} = 1.428$ kOe and $H_{k2} = -5$ kOe. In this critical IP SAF case, when the value of H_{k1} increases, the magnetization configuration will change into T-Type. The corresponding resonance and coupling characteristics of this critical IP SAF are shown in Fig. 1d.” on Page 7, Lines 161-166 in the revised manuscript. This can be understood from Fig. R4, which show the resonance spectra and equilibrium angles of magnetic moments for two H_{k1} values. For $H_{k1} = 1.428$ kOe, as shown in Figs. R4a and R4b, the configuration is IP SAF. With a slight increase in H_{k1} , as shown in Figs. R4c and R4d, the configuration changes into T-Type.

Similarly, we have also changed the statement about the critical PMA SAF case: “Similarly, we consider a critical case for $H_{k1} = -1.428$ kOe and $H_{k2} = 5$ kOe. In this critical PMA SAF case, the magnetization configuration will change into T-Type as the value of H_{k1} decreases. The corresponding resonance and coupling characteristics of this critical IP SAF are shown in Fig. 1e.” on Page 8, Lines 191-194 in the revised manuscript.

Figure R4. **a** and **b** The critical IP SAF case where $H_{k1} = 1.428$ kOe, $H_{k2} = -5$ kOe. **c** and **d** The T-Type where $H_{k1} = 1.429$ kOe, $H_{k2} = -5$ kOe. **(a)** and **(c)** correspond to the calculated resonance spectra, and **(b)** and **(d)** correspond to the calculated angles $\varphi_{1(2)}$ and $\theta_{1(2)}$ as functions of H , respectively.

Comment 2: Regarding the T-type SAFs shown in Fig. 1(f), $H = 0$ Oe was defined as the magnetic field strength at the center frequency, H_0 . But the coupling strengths of the magnon-magnon coupling in T-type SAF should not be defined, because the resonance frequencies of decoupled – and + modes are not crossed, as shown in dashed lines in Fig. 1(f). The coupling strength obtained at such a magnetic field strength are completely meaningless.

Response 2: We agree with the reviewer that the expression of ω_0 in the T-Type configuration in our original version is not appropriate. We thank the reviewer for pointing out this point. In the revised version, we have adopted a more appropriate definition of ω_0 in the T-Type configuration and re-calculated all the theoretical data related to T-Type, which will be mentioned later.

Regarding the coupling strengths, we illustrate the differences between g extracted by quantum models and g extracted by classical models. In most previous works, the magnon-magnon coupling strength g is expressed as half of the gap size, as stated at

the beginning of the theoretical section of our manuscript. However, when the two pure modes do not intersect, a gap already exists between the two pure modes themselves. Therefore, in this situation, half of the gap clearly cannot represent the actual coupling strength, and the g defined by the above classical method also loses its original meaning. This is the reason that previous works rarely discuss such a situation. However, this does not mean that once the two pure modes have different frequencies, there is no interaction between them.

Here, we point out that the coupling strengths corresponding to the above situation can be well defined using a quantum model. The physical quantity of coupling strength is a concept initially derived from cavity quantum electrodynamics and essentially described using quantum models [see Nature Reviews Physics 1, 19-40 (2019)]. The coupling strength obtained from the above classical method is only an approximation under certain conditions, as shown in Section S1 in the Supplementary Materials. In our work, the co-rotating coupling strength g_1 and the counter-rotating coupling g_2 are obtained based on the generalized Hopfield model on the quantum perspective, which can well describe the systems regardless of the value of H . Therefore, g_1 and g_2 are meaningful for different magnetization configurations at any H . And they are actually functions of H , as shown in Figs. 1d to 1f and Figs. 4d to 4f in the revised manuscript. Taking T-Type as an example, we further clarify that g_1 and g_2 are still physically meaningful even when ω_+ and ω_- do not intersect, as shown in Fig. R5. In Fig. R5a (see also Fig. S11b), we show the changes in resonance spectra caused by g_1 and g_2 , respectively. The co-rotating coupling strength $g_1(H)$ causes repulsion between the + and - modes, as shown by the grey dashed curves and red solid curves. The counter-rotating coupling strength $g_2(H)$ causes a redshift of the spectral curves, as shown by the red solid curves and dark solid curves. Therefore, both g_1 and g_2 in T-Type have clear impacts on the resonance spectra. The coupling between the + mode and - mode in T-Type is also reflected in the intensity of the resonance spectra as shown in the micromagnetically simulated Fig. R5b (see also Fig. S4j). The details of micromagnetic simulation can be found in Section S4 in the revised Supplementary Materials. If there

is no coupling, it is expected that only the – (acoustic) mode can be excited. However, in Fig. R5b, both branches are excited, directly indicating the existence of coupling. Therefore, unlike the g obtained by the classical method, g_1 and g_2 defined by the quantum model still have meaning even when the frequencies of the two pure modes are different and can directly lead to various interesting phenomena such as the reduced quantum fluctuations, non-zero magnon numbers in the ground state, and reduced ground state energy, as shown in Figs. R5c to R5e.

Generally, instead of coupling strength, a dimensionless quantity is often preferred to quantify the strength of coupling. The normalized coupling strength g/ω_0 , *i.e.*, the ratio of coupling strength to center frequency ω_0 , is widely adopted. Since g_1 and g_2 in T-Type can be well defined in our quantum model, we also use the normalized g_1 and g_2 in T-Type. On the other hand, we note that T-Type and PMA SAF have many similar characteristics, both of which can be described within the same model. In order to compare g_1 (g_2) of these two magnetization configurations, we mainly focus on the case of $H = 0$ in T-Type. As mentioned on Page 7, Lines 175-176, the center frequency ω_0 indicates the bare energy of the decoupled magnon, *i.e.*, $\omega_0 = \omega_+$ or ω_- . In T-Type, when $H = 0$, since ω_+ and ω_- have different frequencies, we set ω_0 as the larger one of $\omega_+(H = 0)$ and $\omega_-(H = 0)$ in the revised version. In this way, the obtained g_1/ω_0 and g_2/ω_0 will not be exaggerated. Since in our cases, $\omega_-(H = 0) > \omega_+(H = 0)$, the g_1/ω_0 (g_2/ω_0) in T-Type in our work now has a clear meaning: the ratio of g_1 (g_2) to the pure – mode frequency $\omega_- = \omega_0$ when $H = 0$.

We have added the following sentences on Page 9, Lines 209-217 in the revised manuscript as:

“In this case, a significant feature is that the – mode and + mode no longer intersect. Previous works have rarely discussed such a situation because the gap caused by the coupling only accounts for a part of the total gap. However, our systems are described by the quantum model, where g_1 and g_2 can be defined at any H . Therefore, g_1 and g_2 are still meaningful for T-Type. To normalize the coupling strengths, we also define ω_0 in the T-Type case. To avoid exaggerating the effect of coupling and considering that

T-Type and PMA SAF can be described within the same model, we set ω_0 as the larger one of $\omega_+(H_0 = 0)$ and $\omega_-(H_0 = 0)$. In our cases, ω_0 is set to $\omega_-(H_0)$.”

Figure R5. Various effects caused by non-zero coupling strength in T-Type configuration. **a** Calculated branch I and branch II (black solid curves), the decoupled – and + modes (gray dashed curves), and the co-rotating coupled branches where $g_2 = 0$ (red solid curves) of a typical T-Type. See also Fig. S11b. **b** Simulated resonance spectra of a typical T-Type, see also Fig. S4j. **c** Minimum quantum fluctuation as a function of H for a typical T-Type, where a reduced quantum fluctuation is obtained due to the presence of g_2 . See also Fig. S8k. **d** Calculated average + (–) mode magnon numbers as functions of H for a typical T-Type. The non-zero magnon numbers at the ground state indicate the presence of g_2 . See also Fig. S9c. **e** Calculated ground state energy for a typical T-Type. The reduced ground state energy also indicates the presence of g_2 . See also Fig. S12b.

Comment 3: *Is there any effect of the magnetization configuration in PMA and T-type SAFs? For example, in the case of PMA SAF, the magnetic resonance precession directions of low and high frequency modes are reversed in the head-to-head magnetization configuration and tail-to-tail magnetization configuration, as discussed in the previous paper (e.g. Y. Shiota et al., Phys. Rev. Applied 18, 014032 (2022)).*

Response 3: Thanks for the reviewer’s comment and suggestion on the related paper. Y. Shiota *et al.* demonstrated an interesting property in their study: For a PMA SAF system with a magnetic field perpendicular to the sample plane, its resonance features are closely related to the initial configurations, *i.e.*, head-to-head and tail-to-tail

configurations. We have performed micromagnetic simulations and theoretical calculations in our PMA SAF and T-Type systems to see the influence of the initial configurations. We demonstrate that their resonance features are **independent** of the initial configurations when H is applied in the SAF plane for our PMA SAF and T-Type systems. As shown in Fig. R6, we take a typical PMA SAF example to illustrate the effect of different initial configurations on the resonance properties. Figure R6a schematically shows two initial configurations considered for PMA SAF: “head-to-head” and “tail-to-tail”. Head-to-head configuration means $\theta_1 = \theta_2 = 0^\circ$ when $H = 0$, and tail-to-tail configuration means $\theta_1 = \theta_2 = 180^\circ$ when $H = 0$. The resonance spectra of these two configurations are obtained by micromagnetic simulation (color) and theoretical calculation (curves), as shown in Figs. R6b and R6c. From these two plots, it is found that there is no difference in spectra between the two configurations. Figures R6d and R6e show the simulated corresponding equilibrium positions of \mathbf{m}_1 and \mathbf{m}_2 . We use $\theta_{1(H)}$ and $\theta_{2(H)}$ to represent the equilibrium angles obtained in head-to-head configuration, and $\theta_{1(T)}$ and $\theta_{2(T)}$ to represent the equilibrium angles obtained in tail-to-tail configuration. They satisfy the following relations: $\theta_{1(H)} + \theta_{1(T)} = 180^\circ$ and $\theta_{2(H)} + \theta_{2(T)} = 180^\circ$. Figures R6f and R6g further show the magnetization oscillations in the time domain for different initial configurations. For convenience, we define the right-hand (RH) polarization and left-hand (LH) polarization for a single magnetic moment \mathbf{m}_1 (\mathbf{m}_2). The RH (LH) polarization means the magnetic moment precesses counter-clockwise (clockwise) around its equilibrium position, as schematically shown in Fig. R6f. For the case of $H = 1$ kOe, \mathbf{m}_1 and \mathbf{m}_2 process the LH (RH) polarization and RH (LH) polarization when the branch I (II) is excited, respectively. We find that the polarization for a single magnetic moment is independent of the initial configurations. Besides, the amplitude of oscillation of \mathbf{m}_1 (\mathbf{m}_2) is also independent of the initial configurations. For the case of higher H (not shown here), the polarization of \mathbf{m}_1 (\mathbf{m}_2) becomes RH when branch I (II) is excited, which is also independent of the initial configurations. And the amplitude of oscillation of \mathbf{m}_1 (\mathbf{m}_2) is still independent of the initial configurations. We also study the effect of the initial configuration on T-Type

(not shown here), and we find that all the properties mentioned above are satisfied in T-Type, except that only the RH polarization is observed in \mathbf{m}_1 (\mathbf{m}_2). Therefore, their spectra are independent of the initial configurations when H is applied in the SAF plane for both PMA SAF and T-Type structures. And when one of the branches exists, \mathbf{m}_1 (\mathbf{m}_2) of the two initial configurations are symmetric with respect to the x axis (the direction of H) at any time. We point out that this is because H is applied in the plane, and the total effective fields of \mathbf{m}_1 (\mathbf{m}_2) in both initial configurations are equivalent.

In the revised manuscript, we have cited the relevant paper as Ref. 47 and added the following sentences on Pages 10-11, Lines 262-267 in the revised manuscript as:

“In addition, a recent study indicates that for a PMA SAF system with a magnetic field perpendicular to the sample plane, its resonance features are closely related to the initial configurations, i.e., head-to-head and tail-to-tail configurations⁴⁷. While we demonstrate that in our PMA SAF and T-Type systems, their resonance features are independent of the initial configurations when H is applied in the SAF plane, as shown in Section S10.”

The discussion on the effect of the initial magnetization configurations is also shown in Section S10 in the revised Supplementary Materials.

Figure R6. **a** Schematic of micromagnetic simulation configuration. External magnetic field \mathbf{H} is applied in the $+x$ axis. The rf magnetic field is applied in the SAF plane at an angle of 45° (to the $+x$ axis). Two initial configurations: "head-to-head" configuration and "tail-to-tail" configurations are considered in the PMA SAF structure. **b** and **c** Calculated and simulated resonance spectra of a typical PMA SAF example with head-to-head configuration (**b**) and tail-to-tail configuration (**c**). The parameters of this PMA SAF example are: $H_{k1} = 0$ kOe, $H_{k2} = 5$ kOe, and $H_{ex} = -2$ kOe. The color plots show the resonance spectra obtained from micromagnetic simulation, and the violet dashed curves show the resonance spectra obtained from the theoretical calculation. **d** and **e** $\theta_{1(2)}$ as

a function of H obtained from micromagnetic simulation, where (d) and (e) correspond to head-to-head configuration and tail-to-tail configuration, respectively. The insets in each plot show the schematics of equilibrium positions of the magnetic moments \mathbf{m}_1 and \mathbf{m}_2 at different values of H . **f** and **g** Simulated magnetization oscillations of \mathbf{m}_1 and \mathbf{m}_2 when the initial configurations are head-to-head (**f**) and tail-to-tail (**g**). H is set to 1 kOe during the micromagnetic simulation. The schematics of the right-hand (RH) polarization and left-hand (LH) polarization of \mathbf{m}_1 and \mathbf{m}_2 are shown in (**f**). Panels 1 to 4 in each plot show the oscillations of m_x , m_y , and m_z in the time domain, where Panels 1 and 3 correspond to the oscillations of \mathbf{m}_1 (Panel 3) and \mathbf{m}_2 (Panel 1) when the branch II is excited, and Panels 2 and 4 correspond to the oscillations of \mathbf{m}_1 (Panel 4) and \mathbf{m}_2 (Panel 2) when the branch I is excited. The polarization of $\mathbf{m}_{1(2)}$ in each panel is indicated on the right.

Comment 4: *The ST-FMR spectrum obtained in the experiments shown in Fig. 3(b) does not indicate which sample was measured.*

Response 4: Thanks to the reviewer for pointing out our negligence. Since we have re-measured all the samples, Figure 3 has been replaced by the new measurement results. Now, Fig. 3(b) corresponds to **PMA SAF sample S1**. We have added “A typical example of the measured signal of sample S1.” on Page 30, Line 793 in the revised manuscript.

Comment 5: *The spectrum fitting using the multi-Lorentzian functions have been performed. But there is no discussion about the magnon dissipation, excitation efficiency etc. And no spectrum fittings have been done for the other ST-FMR spectra.*

Response 5: Thanks for the reviewer’s comment. In the revised manuscript and Supplementary Materials, we have discussed the magnon dissipation and excitation efficiency. As shown in Fig. R7 (see also Figs. 4d to 4f), we present the g_1 , g_2 extracted from the spectra fitting curves and the dissipation rates κ_h , κ_l in the vicinity of $H = H_0$. Since the magnon dissipations are obtained, the cooperativities can also be calculated. We define the co-rotating (counter-rotating) cooperativity $C_{1(2)}(H) = g_{1(2)}^2(H) / (\kappa_h(H)\kappa_l(H))$ in our systems. Focusing on the case of $H = H_0$, for S3, S1, and S2, $C_1(H_0)$ is determined to be 38.3, 249.5, and 16.1, respectively. And $C_2(H_0)$ is determined to be 59.4, 0, and 106.1, respectively. The large values of $C_{1(2)}(H_0)$ indicate that our systems can effectively transmit information before dissipation.

Figure R7. **a** to **c** Extracted dissipation rates κ_h , κ_l and coupling strengths g_1 , g_2 as functions of H for samples S3 (**a**), S1 (**b**) and S2 (**c**). In each plot, the orange (blue) triangles represent κ_h (κ_l), and dark red (black) curve represents g_1 (g_2). The orange dashed line indicates H_0 . For S3, S1 and S2, g_1/ω_0 are determined to be 0.392, 0.963, 0.232, respectively, while g_2/ω_0 are determined to be 0.487, 0, 0.597, respectively.

Considering the excitation efficiency, due to the complexity of the microwave field sources in ST-FMR, including Oersted field, field-like torque, and damping-like torque, it is difficult to characterize the overall direction of the microwave field. Therefore, we perform micromagnetic simulations to study the excitation efficiency, as shown in Fig. R8 (see also Fig. S4). In this part, we explain the influence of different degrees of asymmetry, magnetization configurations, and field geometries on excitation efficiency. We first consider the IP SAF case, as shown in Figs. R8a to R8d. In this case, we consider two examples corresponding to small and large magnetic anisotropic asymmetries. For the example with small magnetic anisotropic asymmetry shown in Figs. R8a and R8b, a gap is clearly shown, indicating the presence of coupling between the + and - modes. Accompanying the appearance of the gap is the excitation of both branches, even when θ_h is equal to 0° or 90° , as shown by the segments in Figs. R8a and R8b. θ_h represents the angle between the external field \mathbf{H} and rf magnetic field h , and both \mathbf{H} and h are in the SAF plane. Here, for convenience, we define the two branches as the “major branch” and “minor branch” when specifying θ_h as 0° or 90° . The major (minor) branch represents the branch that can (not) be excited for the specified θ_h when the system is symmetric. For example, in IP SAF, when specifying θ_h as 0° , the branch I (II) is the major (minor) branch when $H < H_0$, while the branch II (I) is the major (minor) branch when $H > H_0$. Although the minor branch can be excited in the presence of coupling, the excitation of the minor branch is still weak when the

asymmetry is small, as shown by the segments in the color plots. This feature indicates that the mode hybridization of IP SAF is not sufficient in the case of small magnetic anisotropic asymmetry. For the example with large magnetic anisotropic asymmetry shown in Figs. R8c and R8d, a larger gap is observed. In this case, the excitation of the minor branch is greatly enhanced compared with Figs. R8a and R8b, as shown by the segments in the color plots. This feature indicates that by increasing the degree of magnetic anisotropic asymmetry, the mode hybridization can become more sufficient. We then consider the PMA SAF case, as shown in Figs. R8e to R8h. In this case, similarly, two examples are considered, corresponding to small and large magnetic anisotropic asymmetries, and they have the same degrees of asymmetry $\kappa = (H_{k1} - H_{k2}) / (H_{k1} + H_{k2})$ as the two IP SAF examples, respectively. We find that the PMA SAF case is similar to the IP SAF case in many ways, but there are also differences. For the example with small magnetic anisotropic asymmetry shown in Figs. R8e and R8f, the major branch, is mainly excited in each plot. However, unlike the IP SAF case, the excitation of the minor branch is not weak when H approaches $H_0 = 0$, as shown by the segments in Figs. R8e and R8f. This phenomenon can be explained by the fact that the PMA SAF system has a larger g_1/ω_0 than the IP SAF system when these two systems have the same κ , as discussed in the manuscript. With the improvement of magnetic anisotropic asymmetry in PMA SAF, as shown in Figs. R8g and R8h, the mode hybridization becomes more sufficient, and the excitation of minor branch can be equivalent to or even larger than the major branch. We further decrease H_{k1} to -10 kOe, and the magnetization configuration transfers to T-Type, as shown in Figs. R8i and R8j. In this case, though there is no crossing between the $+$ mode and $-$ mode, we note that the major and minor branches are both excited when $\theta_h = 0^\circ$ or 90° , indicating that the coupling between the $+$ mode and $-$ mode does occur. Besides, the resonance spectra show interesting features. When H is large, the major branch is mainly excited, while the minor branch is mainly excited when H is small.

Figure R8. **a** to **d** Two examples of IP SAF with relatively small (**a** and **b**) and large (**c** and **d**) magnetic anisotropic asymmetries. (**a** and **c**) correspond to the case of $\theta_h = 0^\circ$, and (**b** and **d**) correspond to the case of $\theta_h = 90^\circ$, where θ_h is the angle between the external field \mathbf{H} and rf magnetic field h . For these two IP SAF cases, they share the same H_{k2} (-5 kOe) and H_{ex} (-2 kOe), but have different H_{k1} . In (**a** and **b**), $H_{k1} = -4$ kOe, and in (**c** and **d**), $H_{k1} = 0$ kOe. **e** to **h** Two examples of PMA SAF with relatively small (**e** and **f**) and large (**g** and **h**) magnetic anisotropic asymmetries. Similarly, (**e** and **g**) and (**f** and **h**) correspond to the case of $\theta_h = 0^\circ$ and 90° , respectively. For these two PMA SAF cases, they share the same H_{k2} (5 kOe) and H_{ex} (-2 kOe), but have different H_{k1} : 4 kOe for (**e** and **f**), and 0 kOe for (**g** and **h**). **i** and **j** An example of T-Type, where (**i**) and (**j**) correspond to the case of $\theta_h = 0^\circ$ and 90° , respectively. In this T-Type case, $H_{k1} = -10$ kOe, $H_{k2} = 5$ kOe and

$H_{\text{ex}} = -2$ kOe. In each plot, the panel(s) on the right represent(s) the corresponding curve(s) plotted along the dashed segment(s) on the left panel.

The spectrum fittings have been done to all spectra. Please refer to the response to Comment 6 for details.

Comment 6: *Regarding the experimental results of the color plots in Fig. 4, the resonance peaks (especially the resonance of the high-frequency mode) are unclear, and it is difficult to determine whether they are consistent with the theoretical calculations. It is written as if the coupling strength is obtained experimentally, but in reality, it seems that the coupling strength is only obtained by theoretical or numerical calculations using material parameters estimated from M-H curves and other data.*

Response 6: Thanks for the reviewer's comment. We note that in the original resonance spectra, the signal V_{mix} of the resonance peak largely decreases with the increase of frequency f , which leads to unclear spectra at high frequencies. After investigation, we find that this issue is caused by transmission loss of our ST-FMR measurement system. At high frequencies, the microwave power loss is larger than that at low frequencies. In our original experiments, we do not consider this transmission loss, which results in a significant attenuation of V_{mix} at high frequencies. To solve the problem, we have improved our ST-FMR measurement setup. In the improved method, before ST-FMR measurements, we use the vector network analyzer to characterize the transmission loss $S_{21}(f)$ in our system. During the measurements, we adjust the output power of the signal generator according to $S_{21}(f)$ to compensate for the loss. According to the improved method, we have re-measured all the samples. The comparison between the original resonance spectra and the revised resonance spectra is shown in Fig. R9. As shown in Fig. R9, the quality of the revised resonance spectra can now be guaranteed even at high frequencies.

The Lorentzian functions are adopted to fit the experimental data. If only one branch appears in the spectra or if the two branches can be separately distinguished, we adopt an ordinary single Lorentzian function to extract f_{res} . If the two branches are very close to each other in the spectra, we adopt the multi-Lorentzian function Eq. 10 in the

revised manuscript. We perform the spectrum fittings to all resonance spectra. In Figs. R10a to R10c, we display some typical $V_{\text{mix}}-f$ spectra and the corresponding fitting results. These selected spectra are measured at different H , which can well capture the features of branch I and branch II. The solid curve(s) in each spectrum correspond(s) to the Lorentzian fitting result of experimental data. For IP SAF sample S3, at any H , branch I and branch II can be distinguished separately, and they are far apart in spectra. Therefore, we fit branch I and branch II separately for S3. While for the other two samples, at some H , the branch I and branch II are very close to each other in the spectra. Therefore, the multi-Lorentzian function Eq. 10 is adopted to fit the experimental data for S1 and S2. The spectrum at H_0 for each sample is also displayed. For S3, S1 and S2, $H_0 = 1.9, 0, 0$ kOe, respectively. Since the ST-FMR measurement requires oscillation of resistance due to the AMR (SMR) effect, we actually add a very small H (~ 20 Oe) to the samples for the cases of $H \approx 0$ kOe.

Figure R9. **a to c** The original resonance spectra. **d to f** The revised resonance spectra.

Figure R10. Representative ST-FMR spectra. (a), (b) and (c) correspond to the samples S3, S1 and S2, respectively. In each plot, five representative spectra are shown. The solid curve(s) in each spectrum correspond(s) to the Lorentzian fitting result of experimental data. The blue and violet arrows indicate the resonance frequencies f_r^l and f_r^h , respectively.

In addition to the ST-FMR measurements, as a supplement, we also perform the Brillouin light scattering (BLS) measurements to obtain the resonance spectra. The setup of the BLS measurement is shown in Fig. R11, where we collect backscattered light and the wavelength λ of the incident light is 532 nm. In our BLS experiment, the angle θ_L of incidence is 0, and the wave vector $k = 0$ obtained from the relationship $k = 4\pi \times \sin\theta_L/\lambda$. The applied dc magnetic field \mathbf{H} in the plane is perpendicular to the incidence plane of the light, which corresponds to the Damon-Eshbach spin wave configuration. We display the spectra measured from the BLS method in Fig. R12. In these spectra, both negative frequency peaks and positive frequency peaks can be obtained, where the negative frequency and positive frequency peaks correspond to the Stokes and anti-Stokes modes, respectively. The orange curves correspond to the fitting results, and the violet dashed lines indicate the extracted resonance frequencies.

Figure R11. Schematic of BLS measurement. External magnetic field \mathbf{H} is applied along the $+z$ axis. The light is perpendicular to \mathbf{H} , with an angle of θ_L to the $+y$ axis. See also Fig. S16.

Figure R12. BLS spectra for S3 (a), S1 (b), and S2 (c). The experimental data are fitted through the Lorentzian function, as shown by the orange curves. The violet dashed lines indicate the resonance frequencies extracted from the Stokes peaks and anti-Stokes peaks.

In Fig. R13, we further display the experimentally extracted resonance frequencies from the ST-FMR and BLS measurements as well as the resonance spectra fitted by our theory. Since the Stokes modes and anti-Stokes modes have almost the same resonance frequency, we only show the result of Stokes modes. From these plots, we show that the resonance frequencies obtained by the two techniques are consistent. The fittings can reproduce the experimental data well, which illustrates the reliability of our theory. And the coupling strengths shown in Fig. R7 are obtained from these fitting curves.

Figure R13. Extracted resonance frequencies and the corresponding fitting results for S3 (a), S1 (b), and S2 (c). The blue circles correspond to the resonance frequencies extracted from the ST-FMR spectra. The purple dots, including one in the inset of (a), correspond to the resonance frequencies extracted from the Stokes peaks in Fig. R10. And the orange curves correspond to the fitting results.

In the revised manuscript, we have modified the following sentences:

On Page 11, Lines 286-289: “In addition to the ST-FMR measurements, as a supplement, we also perform the Brillouin light scattering (BLS) measurements to obtain the resonance spectra. The setup of the BLS measurement is shown in Section S13.”

On Page 12, Lines 292-293: “More details of the spectra are shown in Section S14.”

On Page 12, Lines 312-313: “corresponding to the spectra fitting curves. In Figs. 4d to 4f, we present the corresponding g_1 , g_2 and dissipation rates κ_h , κ_l in the vicinity of $H = H_0$.”

On Page 13, Lines 327-333: “We also calculate the cooperativities ... transmit information before dissipation.”

On Page 22, Lines 553-569: “In the other method ... to each other in the spectra,”
Figures 3 and 4 have been revised.

In the revised Supplementary Materials, we have added Section S4, Section S13, Section S14. And we have integrated Figs. R10, R12, and R13 as Fig. S17.

REVIEWERS' COMMENTS

Reviewer #1 (Remarks to the Author):

The authors have fully addressed my concerns. I would like to recommend this paper for publication in Nature Communications.

Reviewer #2 (Remarks to the Author):

The authors have satisfactorily addressed the referees' comments and therefore I recommend it for publication in Nature Communications.

We extend our sincere appreciation to both reviewers for their recognition of our work and for recommending the publication of our manuscript.